# Guided Safe Shooting: model based reinforcement learning with safety constraints

## Abstract

Reinforcement learning (RL) has achieved remarkable results in a variety of complex control tasks and decision-making problems, including the game of Go and autonomous driving. However, applying RL to real-world scenarios, especially those requiring safety-awareness, poses significant challenges. Existing approaches that enforce strict safety guarantees can limit exploration and lead to suboptimal policies in settings where some safety violations are tolerated. Conversely, Quality-Diversity (QD) algorithms maximize exploration and search for high-reward policies, but they require many interactions with the environment, potentially exposing the agent to high-risk situations. In this paper, we propose Guided Safe Shooting (GuSS), a Model-Based RL (MBRL) approach that leverages a QD algorithm as a planner with a soft safety objective. GuSS is the first MBRL approach that combines QD and safety objectives in a principled way. Our experiments, conducted on three OpenAI gym environments with safety constraints, show that GuSS reduces safety violations while achieving higher performances compared to the considered baselines, thanks to its increased exploration.

## 1 Introduction

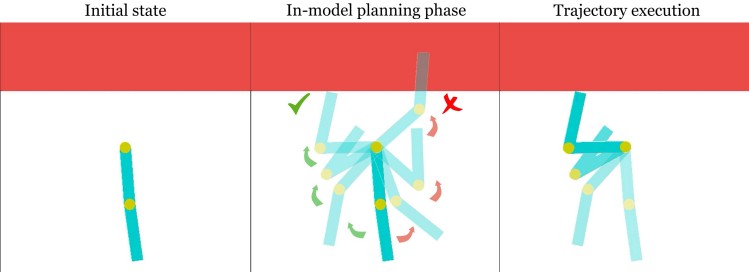

Figure 1: An illustrative example of planning with a model on the Acrobot environment where safety has to be considered. The agent controls the torque on the first joint with the goal of getting its end effector as high as possible, avoiding the unsafe zone (red area). Starting in the rest position (left) the agent uses its model to find the best plan (middle) that will maximize the reward while satisfying the safety constraint and execute it on the real system (right).

Reinforcement Learning (RL) has made significant strides in recent years, solving complex sequential decision-making problems across a range of domains, from controlling industrial robots to mastering the game of Go (Mnih et al., 2015; Andrychowicz et al., 2020; Silver et al., 2016). Despite these successes, the application of RL in real-world systems remains challenging, largely due to safety concerns. The issue lies in the fundamental principle of RL: learning through trial and error to maximize a reward signal (Sutton & Barto, 2018). This approach necessitates unrestricted access to the system for both exploration and action execution, potentially leading to undesirable outcomes and safety risks. For instance, in the task of optimizing a data center's cooling strategy (Lazic et al., 2018), an RL algorithm could inadvertently cause sustained high temperatures,

damaging the system. Safety is paramount in many other applications as well, such as robotics, where unsafe actions can pose risks to both the robot and humans. While such safety violations can be tolerated during training in controlled settings, ensuring safe operation during deployment is crucial. Addressing this issue, known as *safe exploration*, is a central problem in AI safety (Brunke et al., 2022) and is the focus of our research.

In many engineering settings, safety constraints are often defined on two levels: hard and soft (Andersen et al., 2009; Crespo et al., 2006). The hard constraints protect the system from serious damage and are usually enforced by emergency shutdown, which we call a "red-button" scenario. To avoid frequent red-button situations, the system is also equipped with soft safety constraints that steer the agent away from very risky states. While violations of these soft constraints are allowed, even more so at training time, it is important to keep the number of violations as low as possible. Going back to the data center cooling example, the red-button scenario can be one where the servers would be severely damaged. However, one can tolerate soft constraints that are close to the critical temperature to learn to be safe. The learning algorithm should avoid exceeding the soft constraints, but still be flexible enough to cope with sudden increases in temperature due to bursts of computation, without resorting to an emergency shutdown or keeping the temperature too low. Most existing safe-RL approaches tend to strictly enforce any safety constraint, whether hard or soft (Gu et al., 2022; Berkenkamp et al., 2017). While this is useful for critical situations, it can be limiting in the presence of soft-constraints as it can lead to over-cautious solutions that limit exploration and performance.

At the same time, Quality-Diversity (QD) algorithms (Cully & Demiris, 2017), are a family of evolutionary algorithms that maximize exploration while generating a set of high-performing policies. Among them, MAP-Elites (ME) (Mouret & Clune, 2015) is one of the simplest and most powerful version, capable of quickly adapting to new scenarios, such as damages to a robot unseen at training time (Cully et al., 2015), thanks to the extensive exploration performed during training. However, QD requires a high number of interactions with the system and does not take into account any safety constraints, which can be limiting for many real-world scenarios.

In this work, we propose Guided Safe Shooting (GuSS) a novel method that combines QD algorithms with Model-Based RL (MBRL) in the context of soft safety constraints. GuSS utilises ME as a planner in the model to select the best action at decision time, while performing rejection sampling during the optimization performed by ME to take into account any soft constraint encountered during the search. Applying ME in the model provides the agent with the ability to explore a wide range of possible actions, which is crucial in safe RL where a trade-off between reward and safety must be found. Moreover, this reduces the number of interactions with the real system needed by the evolution method, which in turns limits the risk of violating the soft constraints, as most of the violations would happen at planning time in the model itself. This contrasts with many of the methods in the literature that address the problem of finding a safe course of action through Lagrangian optimization or by penalizing the reward function (Webster & Flach, 2021; Ma et al., 2022; Cowen-Rivers et al., 2022). This approach leads to a safer and more efficient search, covering a larger portion of the state space while discovering safer plans.

We evaluate *GuSS* on three different environments: a safe version of Pendulum and Acrobot and SafeCar-Goal from the safety-gym environment (Ray et al., 2019). The experiments demonstrate the ability of the proposed planner to find strategies that achieve high rewards with minimal costs, even in scenarios where these metrics are conflicting, such as the Safe Acrobot environment shown in Fig. 1. Moreover, we evaluate the exploration performance of our agent in an environment in which good exploration is fundamental to safely solve the problem, showing how, thanks to Quality-Diversity (QD), GuSS can generate better and safer plans compared to commonly used approaches.

In summary, the main contribution of this paper is to propose the use of QD methods as planning techniques in safe-MBRL approaches. The advantage of such methods is their increased exploration, which allows to discover many diverse behaviors. Such exploration can be risky in settings where the agent has to be cautious to not damage the system or harm humans. In this paper, we analyze how by taking into consideration the safety cost while exploring in the model, and applying rejection sampling, we can reduce the risk of violating the safety constraints while efficiently solving the task. We do this by selecting the ME algorithm and modifying it to take into account not only the diversity and the quality of the discovered solutions, but

also their safety cost. To our knowledge, this is the first time ME, and QD methods in general, are used as planners for MBRL in a safety-conscious setting.

In the following we will discuss the related works in Sec. 2 and give an overview of the concepts behind safe-RL and QD methods in Sec. 3. We will detail GuSS in Sec. 4 and discuss the results in Sec. 5. The paper concludes with a discussion of possible future research directions Sec. 6.

## 2 Related Work

Some of the most common techniques for addressing safety in Reinforcement Learning (RL) involve solving a Constrained Markov Decision Process (CMDP) (Altman, 1999) through model-free RL methods (Achiam et al., 2017; Ray et al., 2019; Tessler et al., 2018; Hsu et al., 2022; Zhang et al., 2020). A popular approach to solve the CMDP is through Lagrangian-based methods, which transform the constrained optimization problem into an unconstrained form (Ray et al., 2019). Another well-known method is Constrained Policy Optimization (CPO) (Achiam et al., 2017), which adds constraints to the policy optimization process in a way similar to Trust Region Policy Optimization (TRPO) (Schulman et al., 2015). A similar approach is taken by Projected Constrained Policy Optimization (PCPO) (Yang et al., 2020a) and its extension (Yang et al., 2020b). The algorithm works by first optimizing the policy with respect to the reward, then projecting it back onto the constraint set in an iterated two-step process. A different strategy involves storing all the "recovery" actions that the agent took to leave unsafe regions in a separate replay buffer (Hsu et al., 2022). This buffer is then used whenever the agent enters an unsafe state by selecting the most similar transition in the safe replay buffer and performing the same action to escape the unsafe state.

Model-free RL methods need many interactions with the real-system in order to collect the necessary data for training. This can be a significant limitation in situations where safety is critical, as increasing the number of samples could increases the probability of entering unsafe states. Model Based Reinforcement Learning (MBRL) addresses this issue by learning a model of the system that can then be used to learn a safe policy. This allows for increased flexibility in dealing with unsafe situations, particularly when safety constraints change over time. There are several works that use MBRL to tackle safety. A common approach is to rely on Gaussian Processes (GPs) to model the environment and use the dynamics model's uncertainty to guarantee safe exploration (Berkenkamp et al., 2017; Cowen-Rivers et al., 2022). While GPs allow for good representation of the dynamics uncertainty, their usability is limited to low-data, low-dimensional regimes with smooth dynamics (Kuss & Rasmussen, 2003). A common alternative to GPs is the use of ensemble networks (Webster & Flach, 2021; Liu et al., 2020) which scale better. In our work, we use the MBRL setup and choose to use auto-regressive mixture density nets as model which have shown to alleviates error accumulation down the horizon (Kégl et al., 2021) which is an important feature for planning.

The learned dynamics model can then be used to learn a safe policy. In SAMBA (Cowen-Rivers et al., 2022), the authors rely on a modified version of the soft-actor critic algorithm by including the safety constraint with Lagrangian multipliers. Thomas et al. (2021) use an automatic reward shaping approach, which eliminates the need for a separate cost function and falls back into a classical MDP formulation that is solved with SAC trained on the model. A different approach adopted in the control community is to rely on Model Predictive Control (MPC) to select the safest trajectories with a learned or given model in closed-loop fashion (Wen & Topcu, 2018; Liu et al., 2020; Zanon & Gros, 2020). For example, the work of Koller et al. (2018) uses the propagation of uncertainty to recursively guarantee the existence of a safe trajectory that satisfies the constraints of the system. The authors of Uncertainty Guided Cross-Entropy Methods (CEM) (Webster & Flach, 2021) extend PETS (Chua et al., 2018) by modifying the objective function of the CEM-based planner to avoid unsafe areas. In this setting, an unsafe area is defined as the set of states for which the ensemble of models has the highest uncertainty. Vlastelica et al. (2021) relies on CEM to perform MPC zero-order trajectory optimization. The authors use a technique inspired by PETS (Chua et al., 2018) by using predictions particles sampled from the probabilistic models and randomly mixed between ensemble members at each prediction step. In this way, the sampled trajectories are used to perform a Monte Carlo estimate of the expected trajectory cost to estimate the uncertainty of the dynamics. The method we propose draws parallels with the approach by Vlastelica et al. (2021), but diverges in the optimization strategies employed. While the Cross-Entropy Method (CEM) is designed to identify the optimal parameters that

maximize a specific objective function, the Quality-Diversity (QD) Map-Elites algorithm is design to generate a wide array of diverse and high-performing solutions. The advantage of using QD Map-Elites algorithm in generating safe plans is its ability to produce a diverse set of solutions, thereby offering a wider spectrum of safe alternatives rather than concentrating on a singular optimal solution like CEM.

## 3  Background

In this section, we introduce the concepts of safe RL and QD algorithms on which our method builds.

### 3.1  Reinforcement Learning and safe-RL

Episodic RL problems are usually represented as a Markov decision process (MDP) $\mathcal{M} = \langle \mathcal{S}, \mathcal{A}, p_{\text{real}}, R, T \rangle$, where $\mathcal{S}$ is the state space, $\mathcal{A}$ is the action space, $p_{\text{real}} : \mathcal{S} \times \mathcal{A} \rightsquigarrow \mathcal{S}$[1] is the transition dynamics, $R : \mathcal{S} \times \mathcal{A} \rightarrow \mathbb{R}$ is the reward function, and $T \in \mathbb{Z}^+$ is the length of the episode. The state vector $\boldsymbol{s}_t = (s_t^1, \ldots, s_t^{d_{\text{s}}})$ contains $d_{\text{s}}$ numerical or categorical variables, measured on the system at time $t$. The action vector $\boldsymbol{a}_t$ contains $d_{\text{a}}$ numerical or categorical action variables. Given a (deterministic or stochastic) control policy $\pi : \mathcal{S} \rightsquigarrow \mathcal{A}$, we can roll out the policy $\pi$ and the system dynamics $p_{\text{real}}$ to obtain a *trace* or *trajectory* $\mathcal{T} = \big((\boldsymbol{s}_1, \boldsymbol{a}_1), \ldots, (\boldsymbol{s}_T, \boldsymbol{a}_T)\big)$ by repeatedly applying $\boldsymbol{a}_t \leftsquigarrow \pi(\boldsymbol{s}_t)$ and observing $\boldsymbol{s}_{t+1} \leftsquigarrow p_{\text{real}}(\boldsymbol{s}_t, \boldsymbol{a}_t)$. The performance of the policy is measured by the mean reward $\text{MR}(\mathcal{T}) = \frac{1}{T} \sum_{t=1}^T R(\boldsymbol{s}_t, \boldsymbol{a}_t)$. The goal of RL is to find a policy which maximizes the expected mean reward:

$$\pi^* = \underset{\pi}{\arg\max} \, \mathbb{E}_{\mathcal{T} \leftsquigarrow (\pi, p_{\text{real}})} \{\text{MR}(\mathcal{T})\} \tag{1}$$

To incorporate constraints (for example representing safety requirements), we define a cost function $C : \mathcal{S} \times \mathcal{A} \rightarrow \mathbb{R}$ which, in our case, is a simple indicator for whether the system entered into an unsafe state ($C(\boldsymbol{s}, \boldsymbol{a}) = 1$ if the state $\boldsymbol{s}$ is unsafe and $C(\boldsymbol{s}, \boldsymbol{a}) = 0$ otherwise). With this definition, the mean cost $\text{MC}(\mathcal{T}) = \frac{1}{T} \sum_{t=1}^T C(\boldsymbol{s}_t, \boldsymbol{a}_t)$ is an estimate of the probability of unsafe states visited along one trajectory. The new goal is then to find a policy $\pi$ in the policy set $\Pi$ with high expected reward $\mathbb{E}_{\mathcal{T} \leftsquigarrow (\pi, p_{\text{real}})} \{\text{MR}(\mathcal{T})\}$ and low safety cost $\mathbb{E}_{\mathcal{T} \leftsquigarrow (\pi, p_{\text{real}})} \{\text{MC}(\mathcal{T})\}$. One way to solve this new problem is to rely on constrained Markov decision processes (CMDPs) (Altman, 1999) by adding constraints to the expectation (La & Ghavamzadeh, 2013) or to the variance of the return (Chow et al., 2017). CMDP is based on MDP, in which the additional constraint set $\{(\mathcal{C}_i, l_i)\}_{i=1}^k$ is considered, where each $\mathcal{C}_i$ is a cost value function, and $l_i$ the associated safety constraint bound. The feasible policy set $\Pi_c$ that can satisfy the safety constraint bound is as follows:

$$\Pi_c = \cap_{i=1}^k \{\pi \in \Pi \text{ and } \mathcal{C}_i(\pi) \leq l_i\} \tag{2}$$

The new objective to the CMDP can be seen as finding the best policy $\pi$ in the space of considered safe policies $\Pi_c$:

$$\pi^* = \arg\max_{\pi \in \Pi_c} \{\text{MR}(\mathcal{T})\} \tag{3}$$

The hard constraint imposed with the safety constraint bound $l_i$ can be in some applications difficult to tune. When safety is critical (integrity of the system) the safety constraint has to be strict ($l_i = 0$) to ensure no violation at all as it can break the system. For many others applications such critical safety constraint can be relaxed. For example, in a robot navigation task, safety can be defined as the number of collisions. While ideally we would like to have zero collisions, in practice we can accept some violations. Then optimal policy can be rewritten by relaxing the hard constraints on $\mathcal{C}_i$:

$$\pi^* = \arg\min_{\mathcal{C}_i} \max \{\text{MR}(\mathcal{T})\} \tag{4}$$

One requirement for this setting is to ensure that all environment are ergodic MDPs (Hutter, 2002) which guarantee that any state is reachable from any other state by following a suitable policy. In this setup, safety violations can be accepted as at any time-step $t$ it is possible to recover a safe state.

---

[1]We use $\rightsquigarrow$ and $\leftsquigarrow$ to denote both probabilistic and deterministic mapping.

---

**Algorithm 1:** Iterated Batch Model Based RL

---

**1 INPUT:** real-system $p_{\text{real}}$, number of episodes $N$, number of plans for planning step $M$, planning
    horizon length $H$, episode length $T$, initial random policy $\pi^0$
**2 RESULT:** learned model $p(\cdot)$ and planner
**3** $\mathcal{D} = \varnothing$                             `// Initialize empty trace collection`
**4** $\boldsymbol{s}_0 \leftsquigarrow p_{\text{real}}$              `// Sample initial state from real system`
**5 for** $t$ *in [0, ..., T]* **do**
**6**     $\boldsymbol{a}_t \leftsquigarrow \pi^0(\boldsymbol{s}_t)$                       `// Generate random action`
**7**     $\boldsymbol{s}_{t+1} \leftsquigarrow p_{\text{real}}(\boldsymbol{s}_t, \boldsymbol{a}_t)$               `// Apply action on real system`
**8**     $\mathcal{T} = \mathcal{T} \cup (\boldsymbol{s}_t, \boldsymbol{a}_t)$               `// Store transition in trace`
**9** $\mathcal{D} = \mathcal{D} \cup \mathcal{T}$                      `// Store trace in trace collection`
**10 for** $\tau \in [1, ..., N]$ **do**
**11**     $\hat{p}^{(\tau)} \leftarrow \text{TRAIN}(\hat{p}^{(\tau-1)}, \mathcal{D})$          `// Train model on collected traces`
**12**     **for** $t$ *in [0, ..., T]* **do**
**13**        $\boldsymbol{a}_t^{(\tau)} \leftsquigarrow \text{PLAN}(\hat{p}^{(\tau)}, \boldsymbol{s}_t^{(\tau)}, H, M)$      `// Use planner to generate next action`
**14**        $\boldsymbol{s}_{t+1}^{(\tau)} \leftsquigarrow p_{\text{real}}(\boldsymbol{s}_t^{(\tau)}, \boldsymbol{a}_t^{(\tau)})$      `// Apply action on real system`
**15**        $\mathcal{T} = \mathcal{T} \cup (\boldsymbol{s}_t^{(\tau)}, \boldsymbol{a}_t^{(\tau)})$     `// Store transition in trace collection`
**16**     $\mathcal{D} = \mathcal{D} \cup \mathcal{T}$                  `// Store trace in trace collection`

---

## 3.2 MBRL with decision-time planning

In this work, we address safety using an MBRL approach (Moerland et al., 2023) with decision-time planning (Sutton & Barto, 2018), also known as Model Predictive Control (MPC) (Holkar & Waghmare, 2010). This approach approximates the problem in Eq.(3) by repeatedly solving a simplified version of the problem initialized at the currently measured state $\boldsymbol{s}_t$ over a shorter horizon $H$ in a receding horizon fashion. The MPC scheme relies on a sufficiently descriptive transition dynamics of the system to optimize performance and ensure constraint satisfaction. In this setting, the transition dynamics $p_{\text{real}}$ is estimated using the data collected when interacting with the real system. The objective is to learn a model $\hat{p}(\boldsymbol{s}_t, \boldsymbol{a}_t) \rightsquigarrow \boldsymbol{s}_{t+1}$ to predict the next state given the current state and the action and use it to plan to optimize a given performance metric.

In this work, we consider *iterated batch* MBRL (also known as *growing batch* (Lange et al., 2012) or *semi-batch* (Singh et al., 1995)), where the algorithm, illustrate in Alg. 1, starts with an initial random policy $\pi^{(0)}$. Then, in an iteration over $\tau = 1, \ldots, N$, it updates the model $\hat{p}^{(\tau)}$ in a two-step process of (i) performing MPC on the real system $p_{\text{real}}$ for a whole episode to obtain the trace $\mathcal{T}^{(\tau)} = \left((\boldsymbol{s}_1^{(\tau)}, \boldsymbol{a}_1^{(\tau)}), \ldots, (\boldsymbol{s}_T^{(\tau)}, \boldsymbol{a}_T^{(\tau)})\right)$, (ii) training the model $\hat{p}^{(\tau)}$ on the growing collection of transition data $\mathcal{D} = \bigcup_{\tau'=1}^{\tau} \mathcal{T}^{(\tau')}$ collected up to iteration $\tau$ [2]. The process is repeated until a given number of evaluations $N$ or a target performance are reached. While the system model is a core element of MPC, having a good controller (i.e. policy) also has a major influence on the resulting closed-loop performance. One way to see the MPC controller is as a search problem where at each time-step $t$ we produce a set of possible policies $\Pi_M$ and evaluate them according to our performance criteria (in our case reward and safety). This is possible because it is cheap to evaluate each policy with $r$ and $c$ and rank them according to our criterion. With a good planner that spans the search space we can expect to have $\Pi_c \subseteq \Pi_M$, then the optimal policy $\pi^*$ can be found by taking the objective of equation (4) such that there is no other policy $\pi' \in \Pi_M$ with a lower $\text{MC}(\mathcal{T})$ and higher $\text{MR}(\mathcal{T})$ (Ray et al., 2019).

## 3.3 Quality-Diversity and MAP-Elites

Quality-Diversity methods belong to the family of Evolution Algorithms (EAs) and are designed to achieve two goals simultaneously: generating policies that exhibit diverse behaviors and achieving high performance

---

[2]In order to keep notation light, we will omit the superscript $^{(\tau)}$ when it is clear to which episode the tuple $(\boldsymbol{s}_1^{(\tau)}, \boldsymbol{a}_1^{(\tau)})$ belongs.

(Pugh et al., 2016; Cully & Demiris, 2017). Each policy is parametrized by $\phi_i \in \Phi$ and is executed on the system, resulting in a trajectory $\mathcal{T}_i$. The trajectory is then mapped to a *behavior descriptor* $b_i \in \mathcal{B}$ through an associated behavior function: $f(\mathcal{T}_i) = b_i \in \mathcal{B}$. The space $\mathcal{B}$ is a hand-designed space in which the behavior of each policy is represented. By maximizing the distance of the policies in this space, QD methods can generate a collection of highly diverse policies. Note that the choice of $\mathcal{B}$ is fundamental and is dependent both on the environment and on the kind of task the agent has to solve in the environment. For example, in a maze navigation task, the behavior descriptor could be the position reached by the robot at the end of the episode, or in a robotic walking task the percentage of time each foot touches the ground (Paolo et al., 2020; Cully et al., 2015). While some works introduced approaches to autonomously learn a representation of the behavior space (Paolo et al., 2020; Cully, 2019; Paolo et al., 2023), in our work we hand-design $\mathcal{B}$ to better analyze the safety performance of the proposed planner. The policies discovered during the search are then optimized with respect to the reward according to different strategies depending on the algorithm (Mouret & Clune, 2015; Paolo et al., 2021; Fontaine & Nikolaidis, 2021).

---

**Algorithm 2:** MAP-Elites

---

1 **INPUT:** real system $p_{\text{real}}$, initial state $s_0$, total evaluated policies $M$, parameter space $\Phi$, discretized behavior space $\bar{\mathcal{B}}$, number of initial policies $n$, mutation parameter $\Sigma$, episode length $T$;

2 **RESULT:** collection of policies $\mathcal{A}_{\text{ME}}$;

3 $\mathcal{A}_{\text{ME}} = \varnothing$ ;                                   // Initialize empty collection

4 $\Gamma \leftarrow \text{SAMPLE}(\Phi, n)$ ;                              // Sample initial $n$ policies

5 **for** $\phi_i \in \Gamma$ **do**

6      $r_i, \mathcal{T}_i = \text{ROLLOUT}(p_{\text{real}}, \boldsymbol{s}_0, \phi_i, h)$ ;                   // Evaluate policy on the model

7      $\bar{b}_i = f(\phi_i, \mathcal{T}_i)$ ;                         // Calculate policy behavior descriptors

8      $\mathcal{A}_{\text{ME}} \leftarrow \text{STORE}(\mathcal{A}_{\text{ME}}, \phi_i, \bar{b}_i, r_i, \bar{\mathcal{B}})$ ;                   // Update collection

9 **while** $M$ *not depleted* **do**

10      $\Gamma \leftarrow \text{SELECT}(\mathcal{A}_{\text{ME}}, K)$ ;            // Randomly sample $K$ policies from collection

11      **for** $\phi_i \in \Gamma$ **do**

12          $\phi' = \phi_i + \epsilon, \ \ \text{with} \ \ \epsilon \sim N(0, \Sigma)$ ;                   // Add noise to policy parameters

13          $r', \mathcal{T} = \text{ROLLOUT}(p_{\text{real}}, s_t, \phi', h)$ ;                   // Evaluate policy on the model

14          $\bar{b}' = f(\phi', \mathcal{T})$ ;                        // Calculate policy behavior descriptors

15          $\mathcal{A}_{\text{ME}} \leftarrow \text{STORE}(\mathcal{A}_{\text{ME}}, \phi', \bar{b}', r', \bar{\mathcal{B}})$ ;                   // Update collection

---

### 3.3.1 MAP-Elites

In this work, we select the MAP-Elites (ME) algorithm (Mouret & Clune, 2015) from the range of QD methods (Lehman & Stanley, 2011; Mouret & Clune, 2015; Paolo et al., 2021) due to its simplicity and effectiveness. MAP-Elites (ME) operates by discretizing the behavior space $\mathcal{B}$ into a grid $\bar{\mathcal{B}}$ and searching for the best policies whose discretized behaviors fill up the cells of the grid. The algorithm starts by sampling the parameters $\phi \in \Phi$ of $n$ policies from a random distribution and evaluating them in the environment. The behavior descriptor $b_i \in \mathcal{B}$ of a policy $\phi_i$ is then calculated from the trace $\mathcal{T}_i$ generated during the policy evaluation. This descriptor is then assigned to the corresponding cell in the discretized behavior space. If no other policy with the same behavior descriptor has been discovered previously, $\phi_i$ is stored as part of the collection of policies $\mathcal{A}_{\text{ME}}$ returned by the method. On the contrary, if another policy with the similar discretized descriptor is already present in the collection, only the one with the highest reward is kept. This operation is performed by the STORE function shown in Alg. 3 and allows the gradual increase of the quality of the policies stored in $\mathcal{A}_{\text{ME}}$. At this point, ME randomly samples a policy from the collection, and uses it to generate a new policy $\tilde{\phi}_i$ to evaluate by adding random noise to its parameters. The cycle repeats until the given evaluation budget $M$ is depleted. The pseudo-code of ME is shown in Alg. 2, while the STORE function, used in lines 8 and 15 of Alg. 2, is shown in Alg. 3.

---

**Algorithm 3:** STORE function of MAP-Elites

---

**1** **INPUT:** collection of policies $\mathcal{A}_{\mathrm{ME}}$, policy parameters $\phi$, discretized policy behavior descriptor $\bar{b}$, policy
   reward $r$, discretized behavior space $\bar{\mathcal{B}}$

**2** **RESULT:** updated policy collection $\mathcal{A}_{\mathrm{ME}}$

**3** **if** $\bar{\mathcal{B}}[\bar{b}] = \varnothing$ **then**
   /* If no policy with similar behavior descriptor has been found            */

**4**   $\mathcal{A}_{\mathrm{ME}} = \mathcal{A}_{\mathrm{ME}} \cup (\phi, \bar{b}, r)$ ;                                 // Add policy to collection

**5** **else**

**6**   $(\phi', \bar{b}', r') \leftarrow \mathcal{A}_{\mathrm{ME}}[\bar{b}]$ ;             // Get policy with similar behavior from collection

**7**   **if** $r > r'$ **then**

**8**     $\mathcal{A}_{\mathrm{ME}} = \left(\mathcal{A}_{\mathrm{ME}} - (\phi', \bar{b}', r')\right) \cup (\phi, \bar{b}, r)$ ;        // Store the new policy with higher reward

---

## 4 Guided safe shooting

This section explains in detail how GuSS works by first describing the training of the learned model and then how the safe-planning process is carried by the QD planner, where we highlight the differences with the base ME algorithm.

### 4.1 The learned system model

The goal of model learning in MBRL is, in each iteration $\tau$, to learn $\hat{p}^{(\tau)} : (\boldsymbol{s}_t, \boldsymbol{a}_t) \rightsquigarrow \boldsymbol{s}_{t+1}$ from the collected traces $\mathcal{D} = \bigcup_{\tau'=1}^{\tau} \mathcal{T}^{(\tau)}$ (Alg. 1). As model class we use autoregressive (DARMDN) and non-autoregressive mixture density nets (DMDN) (Bishop, 1994), that have recently been used in multiple works (Kégl et al., 2021; Chua et al., 2018; Wang et al., 2019). Both are neural nets, outputting parameters of Gaussian distributions, conditioned on the previous state and action. In DARMDN, we learn $d_{\mathrm{s}}$ autoregressive deep neural nets, where $p_0(s_{t+1}^0 | \boldsymbol{s}_t, \boldsymbol{a}_t)$ and $p_\ell(s_{t+1}^\ell | s_{t+1}^0, \ldots, s_{t+1}^{\ell-1}, \boldsymbol{s}_t, \boldsymbol{a}_t)$, $\ell = 1, \ldots, d_{\mathrm{s}-1}$, outputting a scalar mean and standard deviation for each dimension of the state vector. DMDN learns a single spherical $d_{\mathrm{s}}$-dimensional Gaussian, outputting a mean vector and a standard deviation vector. Both models are trained to maximize the likelihood on the training data $\mathcal{D}$. We choose DARMDN for smaller dimensional systems and DMDN for SafeCar-Goal. The models are trained by optimizing the negative log likelihood loss:

$$\mathcal{L} = \mathbb{E}_{\boldsymbol{s}, \boldsymbol{a}, \boldsymbol{s}' \sim \mathcal{T}} \left\{ -log\mathcal{N}\left(\boldsymbol{s}'; \boldsymbol{s} + \mu_\theta(\boldsymbol{s}, \boldsymbol{a}), \textstyle\sum(\boldsymbol{s}, \boldsymbol{a})\right) \right\}, \tag{5}$$

where $(\boldsymbol{s}, \boldsymbol{a}) = (\boldsymbol{s}_t, \boldsymbol{a}_t)$ and $\boldsymbol{s}' = \boldsymbol{s}_{t+1}$. All hyperparameters were tuned on static data generated from a random policy and kept unchanged for all episodes.

### 4.2 Model-based safe quality-diversity

The main contribution of this paper is the application of QD as a planner for a MBRL algorithm in the context of safe-RL with soft safety constraints. This enables the use of this powerful exploration technique in scenarios where system access is costly in terms of resources and safety. To find good policies, QD algorithms need to perform a lot of exploration. In real settings this can be costly and dangerous, as it can lead the system in unsafe states. By using a learned model, the algorithm can be used as a planner in the model, allowing great exploration at minimum cost and risk. In our approach, the QD planner (Alg. 4) is invoked by GuSS at each time step $t$ to generate the action $\boldsymbol{a}_t$ for the system (Line 13 in Alg. 1). The planner begins by initializing a pool of $n$ planning policies, denoted as $\Gamma = \{\phi_i\}_{i=1}^n$ (Line 4 Alg. 4).

In our case, these policies are represented by neural networks with random weights. These policies are evaluated on the learned model $\hat{p}$ for a duration of $H$ timesteps, starting from the current state of the real system $\boldsymbol{s}_t$. The evaluation process produces a reward $r_i$, a trace of simulated states $\tilde{\mathcal{T}}i = ((\boldsymbol{s}_t, \boldsymbol{a}_0), \ldots, (\boldsymbol{s}_{t+h}, \boldsymbol{a}_h))$, and a cost $c_i = \sum_{j=0}^h C(\boldsymbol{s}_{t+j}, \boldsymbol{a}_j)^i$ for each policy $\phi_i$, where $C(\boldsymbol{s}_{t+j}, \boldsymbol{a}_j)$ is the cost function defined in Sec. 3.1. To map the trace to the policy's discretized behavior descriptor, we employ a hand-designed, environment-specific behavior function $f(\cdot)$, resulting in the discretized representation $\bar{b}_i \in \bar{\mathcal{B}}$. Finally, the evaluated policies are

---

**Algorithm 4:** Safety-aware QD planner

---

1 **INPUT:** model $\hat{p}$, current real-system state $\boldsymbol{s}_t$, planning horizon $H$, evaluated action sequences $M$, discretized behavior space $\bar{\mathcal{B}}$, number of initial policies $n$, mutation parameter $\Sigma$, number of policies per iteration $n_{new}$, policy parameter space $\Phi$

2 **RESULT:** action to perform $\boldsymbol{a}_t$

3 $\mathcal{A}_{\text{ME}} = \varnothing$         `// Initialize empty collection of policies`

4 $\Gamma \leftarrow \text{SAMPLE}(\Phi, n)$         `// Sample initial n policies from Φ`

5 **for** $\phi_i \in \Gamma$ **do**

6    $r_i, c_i, \tilde{\mathcal{T}}_i = \text{ROLLOUT}(\hat{p}, \boldsymbol{s}_t, \phi_i, h)$         `// Evaluate policy on the model`

7    $\bar{b}_i = f(\phi_i, \tilde{\mathcal{T}}_i)$         `// Calculate policy behavior descriptors`

8    $\mathcal{A}_{\text{ME}} \leftarrow \text{STORE}(\mathcal{A}_{\text{ME}}, \phi_i, \bar{b}_i, r_i, c_i, \bar{\mathcal{B}})$         `// Update collection`

9 **while** *M not depleted* **do**

10    $\Gamma \leftarrow \text{SELECT}(\mathcal{A}_{\text{ME}}, n_{new}, \Phi)$         `// Select n_new policies`

11    **for** $\phi_i \in \Gamma$ **do**

12       $\phi' = \phi_i + \epsilon, \ \ \text{with} \ \ \epsilon \sim N(0, \Sigma)$         `// Add noise to policy parameters`

13       $r', c', \tilde{\mathcal{T}} = \text{ROLLOUT}(\hat{p}, s_t, \phi', h)$         `// Evaluate policy on the model`

14       $\bar{b}' = f(\phi', \tilde{\mathcal{T}})$         `// Calculate policy behavior descriptors`

15       $\mathcal{A}_{\text{ME}} \leftarrow \text{STORE}(\mathcal{A}_{\text{ME}}, \phi', \bar{b}', r', c', \bar{\mathcal{B}})$         `// Update collection`

16 $\Gamma_{\text{lc}} \leftarrow \mathcal{A}_{\text{ME}}[\min c]$         `// Get policies with lowest cost`

17 $\phi_{\text{best}} \leftarrow \Gamma_{\text{lc}}[\max r]$         `// Get policy with highest reward`

18 $a_t \looparrowleft \phi_{\text{best}}(s_t)$         `// Get next action`

---

stored in the collection $\mathcal{A}_{\text{ME}}$ through the STORE function (Lines `5-8` in Alg. 4). The STORE function (Alg. 5) is a fundamental part of the planner as, contrary to what is done in the vanilla ME STORE function (Alg. 3), it is here that the policies' safety comes into play in the evaluation. During the storage process, a policy can be handled in two possible ways. When a policy $\phi_i$ with a unique discrete behavior descriptor is encountered, the tuple $(\phi_i, \bar{b}_i, c_i, r_i)$ is directly added to the collection, *independently of its cost or reward*. However, if another policy $\phi_j$ with $\bar{b}_j = \bar{b}_i$ already exists in the collection, only the better of the two policies is retained. Note that in this context the "better policy" refers to the one with the *lowest cost*. In the case of the two policies having the same cost, the one with the highest reward is kept. This strategy allows the generation of a low-cost and high-reward collection of policies that can then be used to generate the next action at the end of the planning episode.

---

**Algorithm 5:** STORE function of safety-aware planner

---

1 **INPUT:** collection of policies $\mathcal{A}_{\text{ME}}$, policy parameters $\phi$, discretized policy behavior descriptor $\bar{b}$, policy reward $R$, policy cost $C$, discretized behavior space $\bar{\mathcal{B}}$

2 **RESULT:** updated policy collection $\mathcal{A}_{\text{ME}}$

3 **if** $\bar{\mathcal{B}}[\bar{b}] = \varnothing$ **then**

    `/* If no policy with similar behavior descriptor has been found        */`

4    $\mathcal{A}_{\text{ME}} = \mathcal{A}_{\text{ME}} \cup (\phi, \bar{b}, C, R)$ ;         `// Add policy to collection`

5 **else**

6    $(\phi', \bar{b}', C', R') \leftarrow \mathcal{A}_{\text{ME}}[\bar{b}]$ ;         `// Get policy with similar behavior from A_ME`

7    **if** $(C < C')$ *or* $(C = C' \ \& \ R > R')$ **then**

8       $\mathcal{A}_{\text{ME}} = \big(\mathcal{A}_{\text{ME}} - (\phi', \bar{b}', C', R')\big) \cup (\phi, \bar{b}, C, R)$ ;         `// Replace φ' with φ in A_ME`

---

The planner then starts an iteration of multiple evolutionary generations until a total of $M$ planning policies have been evaluated (Lines `9-15` Alg. 4). In each generation, a pool $\Gamma$ of $n_{new}$ policies with cost $c = 0$ are uniformly sampled from $\mathcal{A}_{\text{ME}}$ through the SELECT function (Alg. 6). If not enough policies with zero cost are present in the collection, additional policies with random weights are included in $\Gamma$ until its size

matches $n_{new}$ (Line 5 Alg. 6). This is different than what done in vanilla ME, as no safety constraints are considered there, but facilitates increased exploration, aiding in the discovery of safer planning policies. The parameters of the policies in $\Gamma$ are then perturbed by adding Gaussian noise $\epsilon \sim N(0, \Sigma)$ to generate new policy parameters, denoted as $\phi' = \phi_i + \epsilon$. The new policies are evaluated on the model and stored in the collection using the STORE function (Lines 11-15 Alg. 4). The genetic perturbation and selection of the policies allows to continually refine the collection until a total of $M$ policies have been evaluated. At this point, a pool $\Gamma_{lc}$ consisting of policies with the lowest cost is selected from $\mathcal{A}_{\mathrm{ME}}$. Among these policies, the one with the highest reward is chosen as the final policy to generate the next action for application on the real system, i.e., $\boldsymbol{a}_t = \phi_{\mathrm{best}}(\boldsymbol{s}_t)$ (Lines 16-18 Alg. 4).

---

**Algorithm 6:** SELECT function of safety-aware planner

---

1 **INPUT:** collection of policies $\mathcal{A}_{\mathrm{ME}}$, number of selected policies $n_{new}$, policy parameter space $\Phi$
2 **RESULT:** set of selected policies $\Gamma$
3 $\Gamma = \mathcal{A}_{\mathrm{ME}}[n_{new}, C = 0]$            // Select $n_{new}$ policies with $C = 0$ from collection
4 **if** $|\Gamma| < n_{new}$ **then**
5     $\Gamma = \Gamma \cup \mathrm{SAMPLE}(\Phi, |\Gamma| - n_{new})$           // Sample missing policies from $\Phi$

---

## 5 Experiments

### 5.1 Safe exploration with known model of environment

In this section, we test our safety-aware planner on a toy environment (Fig. 2.(a)) to highlight the importance of exploration in tasks with safety constraints and analyze the performance of our algorithm. The environment is designed to require a significant amount of safe exploration to reach the goal. The agent must move from Start to the Goal by observing its current position, $\boldsymbol{s} = (x, y)$, and performing actions, $\boldsymbol{a} = (\delta_x, \delta_y) \in \{-1, 0, 1\}$, which control the $(x, y)$ movement at the next time step. At each time step, the reward is given by the negative distance between the agent and the goal, $R(\boldsymbol{s}, \boldsymbol{a}) = -\|\boldsymbol{s} - \boldsymbol{s}_{goal}\|^2$, while the cost is equal to $C(\boldsymbol{s}, \boldsymbol{a}) = 1$ if the agent is in the unsafe areas and to $C(\boldsymbol{s}, \boldsymbol{a}) = 0$ otherwise. Each algorithm has an evaluation budget of $N = 500$ plans with a planning horizon $H = 50$. The behavior space used by GuSS is the plans' final $(x, y)$ position, while the mutation parameter is set to $\Sigma = 0.05$. We compare our method against three other planners: CEM (Chua et al., 2018), RCEM (Liu et al., 2020) and Safe Random Shooting (S-RS). The first one does not take into account the safety constraints while the other two do, with S-RS being a random-shooting planner that rejects all plans with safety violations.

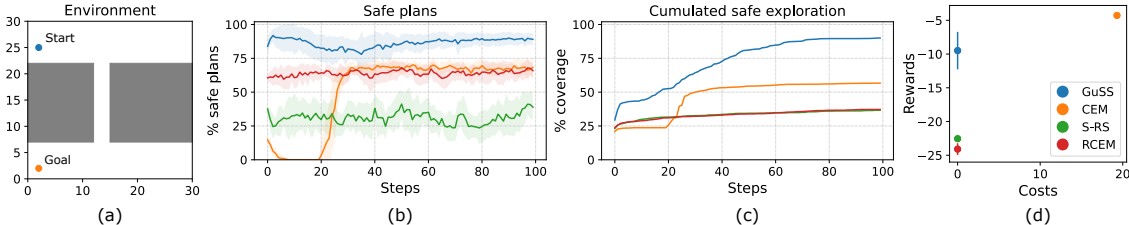

Figure 2: (a) Toy environment. The agent has to navigate from Start to Goal without traversing the unsafe areas in gray. (b) Percentage of generated safe plans at each step. (c) Total amount of the safe space explored through safe plans. (d) Average performance of the algorithms. The results show the mean over 10 seeds, shaded areas represent one standard deviation.

To decouple the performance of the planners from the performance of the model, we perform no model learning and instead use a perfect model to evaluate the plans. The evaluation is performed for a single episode of length $T = 100$ over 10 random seeds. The exploration is evaluated as the percentage of safe plans generated at each step (Fig. 2.(b)) and the amount of safe space covered by the generated safe plans (Fig. 2.(c)). This last metric is calculated by dividing the space into a $50 \times 50$ grid and counting the percentage of

cells in the safe space visited by *safe plans*. We can see that GuSS performs better than the other approaches in both metrics ($p < 1.8e - 4$), generating a high percentage of safe plans while exploring a large portion of the safe state-space. This advantage is also reflected in the high reward with zero cost that GuSS can obtain (Fig. 2.(d)). At the same time, the two other safe planners (RCEM and S-RS) explore a very low percentage of the safe space, never discovering the narrow path that leads to the goal. The only other planner obtaining high rewards is the unsafe CEM. This is achieved by directly traversing the unsafe areas, as can be seen by the percentage of safe plans dropping to 0% in Fig. 2.(b) before going up again once the agent leaves the unsafe zone. A representation of the trajectories generated by the tested planners is shown in Appendix B.

## 5.2 Safe exploration with world model learning

We evaluated the performance of GuSS on three different OpenAI gym environments with safety constraints: pendulum swing-up, Acrobot with discrete actions and SafeCar-Goal from the safety-gym environment (Ray et al., 2019). In designing the environments, we followed previous work (Cowen-Rivers et al., 2022; Ray et al., 2019) with the exception of SafeCar-Goal, for which we used the original version by Ray et al. (2019) with the position of the unsafe areas randomly resampled at the beginning of each episode. Moreover, we use lidar observations rather than the robot position. Each environment can be seen as an ergodic MDP so at each safe time-step it is possible to recover a safe policy (Hutter, 2002). A detailed description of the environments and the hyperparameters optimized by grid search are provided in Appendix A.

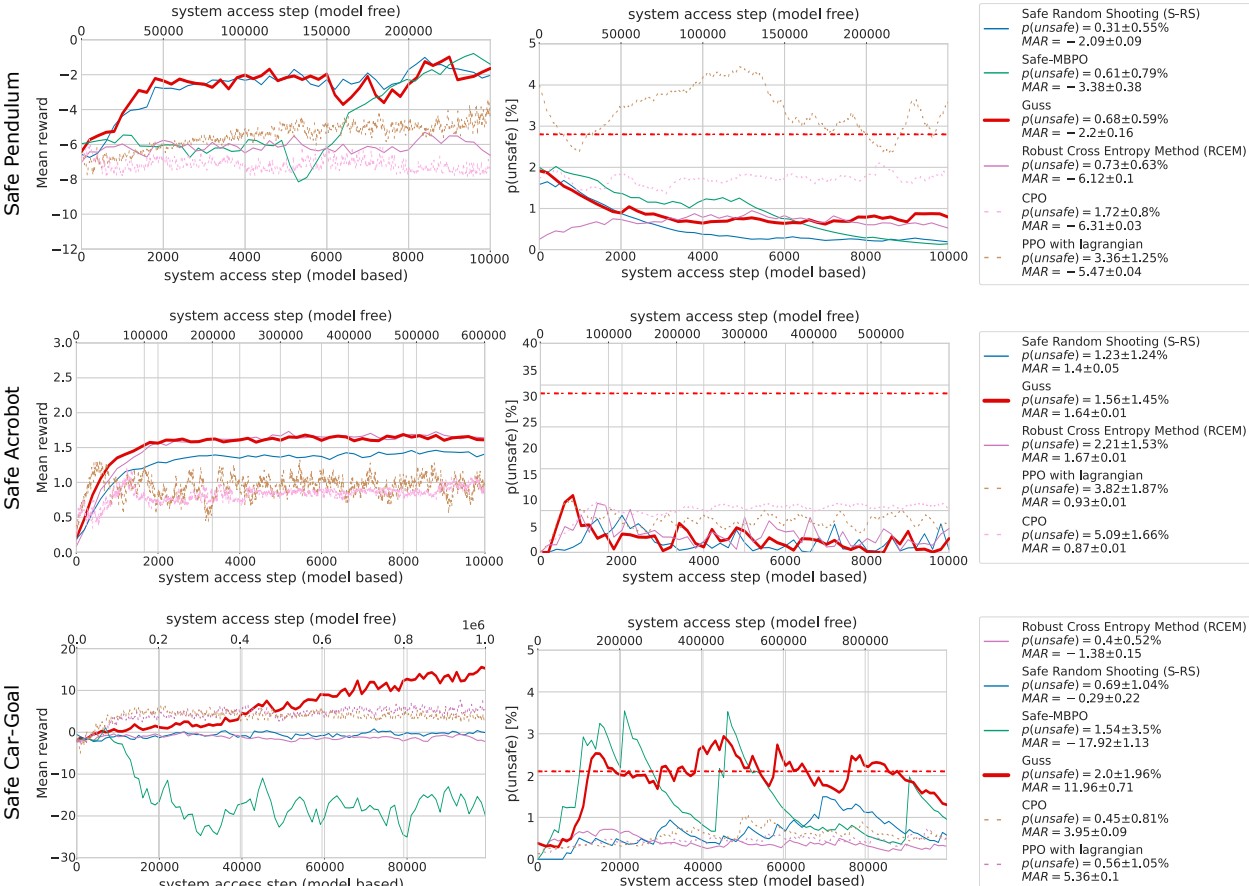

Figure 3: Mean reward and probability percentage of unsafety for the three test environments. Dashed curves indicate Model-free baselines and plain one Model-based approach. The red dashed line indicates the random unsafe probability. All curves represent the mean over 6 random seed.

Table 1: Summary of the different methods on the three different environments. All the metrics are average over all epochs and seeds and ↓ and ↑ mean lower and higher the better, respectively. The best **safe methods** with respect to each metric are highlighted in bold. All ± values are 90% Gaussian confidence interval.

| Method | MAR ↑ | | MRCP×$10^3$ ↓ | | $p$(unsafe)[%] ↓ | | $p$(unsafe)$_{trans}$[%] ↓ |
|---|---|---|---|---|---|---|---|
| | | | Safe Pendulum $r_{thr}$ = −2.5 | | | | |
| S-RS | **-2.09** | **± 0.09** | 2.07 | ± 1.21 | **0.31** | **± 0.55** | 1.11 ± 1.12 |
| GuSS | -2.2 | ± 0.16 | **1.7** | **± 0.88** | 0.68 | ± 0.59 | **0.68 ± 0.65** |
| Safe-MBPO | -3.38 | ± 0.38 | 7.1 | ± 0.61 | 0.61 | ± 0.79 | 1.56 ± 0.83 |
| RCEM | -6.12 | ± 0.10 | - | ± - | 0.73 | ± 0.63 | 0.94 ± 0.52 |
| RS | -2.7 | ± 0.14 | 2.43 | ± 1.03 | 1.49 | ± 0.96 | 1.26 ± 0.49 |
| CEM | -2.99 | ± 0.15 | 1.57 | ± 0.36 | 1.57 | ± 0.80 | 1.67 ± 0.43 |
| CPO | -6.31 | ± 0.03 | 22 | ± 0.0 | 1.72 | ± 0.80 | 1.61 ± 0.72 |
| PPO lag | -5.47 | ± 0.04 | 138 | ± 34 | 3.36 | ± 1.25 | 2.63 ± 0.85 |
| | | | Safe Acrobot $r_{thr}$ = 1.6 | | | | |
| S-RS | 1.4 | ± 0.05 | 1.6 | ± 0.21 | **1.23** | **± 1.24** | **0.85 ± 1.09** |
| GuSS | 1.64 | ± 0.01 | **1.36** | **± 0.25** | 1.56 | ± 1.45 | 3.24 ± 2.51 |
| RCEM | **1.67** | **± 0.01** | 1.6 | ± 0.37 | 2.21 | ± 1.53 | 2.03 ± 1.73 |
| RS | 2.06 | ± 0.01 | 1.33 | ± 0.25 | 20.94 | ± 8.86 | 4.08 ± 3.61 |
| CEM | 2.09 | ± 0.02 | 1.40 | ± 0.43 | 20.07 | ± 9.11 | 3.00 ± 2.90 |
| CPO | 0.87 | ± 0.01 | 87 | ± 59 | 5.09 | ± 1.66 | 3.83 ± 2.20 |
| PPO lag | 0.93 | ± 0.01 | 26 | ± 4 | 3.82 | ± 1.87 | 4.37 ± 2.19 |
| | | | SafeCar-Goal $r_{thr}$ = 10 | | | | |
| S-RS | -0.29 | ± 0.22 | - | ± - | 0.69 | ± 1.04 | 0.49 ± 0.83 |
| GuSS | **11.96** | **± 0.71** | **48.17** | **± 13.69** | 2. | ± 1.96 | 2.17 ± 2.44 |
| Safe-MBPO | -17.92 | ± 1.13 | - | ± - | 1.54 | ± 3.50 | 2.93 ± 3.29 |
| RCEM | -1.38 | ± 0.15 | - | ± - | **0.40** | **± 0.52** | 0.60 ± 0.58 |
| RS | -0.43 | ± 0.18 | - | ± - | 0.63 | ± 1.09 | 0.51 ± 0.65 |
| CEM | 3.87 | ± 0.72 | 60 | ± 6.41 | 1.49 | ± 1.89 | 1.11 ± 1.49 |
| CPO | 3.95 | ± 0.09 | 246 | ± 136.4 | 0.45 | ± 0.81 | 0.37 ± 0.52 |
| PPO lag | 5.36 | ± 0.1 | 256 | ± 87 | 0.56 | ± 1.05 | **0.35 ± 0.43** |

### 5.3 Results

We compared GuSS against various baselines to determine how much different the performances of safe MPC methods are in comparison to unsafe ones. We compared against **RS** (Nagabandi et al., 2018) and its safe version S-RS planner, **CEM** (Chua et al., 2018), and two recent safe MBRL approaches: **Safe-MBPO** (Thomas et al., 2021), and **RCEM** (Liu et al., 2020) . Additionally, to demonstrate the sample efficiency of model-based approaches, we compared against two safe model-free baselines: **CPO** (Achiam et al., 2017), **PPO Lagrangian**; all of which come from the Safety-Gym benchmark (Ray et al., 2019).

The algorithms were compared according to four metrics: Mean Asymptotic Reward (**MAR**), Mean Reward Convergence Pace (**MRCP**), Probability percentage of unsafety (**p(unsafe)[%]**) and transient probability percentage of unsafety (**p(unsafe)[%]**$_{trans}$). The details on how these metrics are calculated are defined in the Appendix D. The MAR scores and the $p$(unsafe)[%] for the pendulum system, Acrobot system and SafeCar-Goal environment are shown in Fig. 3 and in Table 1. The results on the three environments show that the increased exploration provided by GuSS allows it to solve the task without incurring in high costs. On the pendulum system, while Safe-MBPO ($p < 5e - 7$) reaches the highest reward, it needs many more

episodes than GuSS. At the same time, GuSS and S-RS reach similar high MAR ($p = 7.43e - 2$) and low cost ($p = 1.64e - 1$). The other approaches (RCEM, CPO and PPO langrangian) have instead much lower rewards than GuSS ($p < 5.5e - 17$), and fail at solving this simple environment. As expected, on the Acrobot system, safe methods cannot reach the highest MAR scores possible due to the safety constraints blocking the high-rewarding states. GuSS outperforms the simple S-RS method in terms of MAR ($p < 3e - 6$) but has a slightly lower MAR than RCEM ($p < 1e - 4$). At the same time, the three algorithms have similar low cost ($p > 0.5$). On this environment as well, the two model-free approaches reach lower MAR scores than the model-based ones ($p < 7e - 14$) with higher costs ($p < 8e - 3$) and a much higher number of system access steps. Note that Safe-MBPO has not been tested on this environment as Safe-MBPO's SAC only works on continuous action spaces. The advantage of using a QD-based planner compared to simpler ones as S-RS and RCEM is clear from the results on the hardest of the three environment: SafeCar-Goal. GuSS is the only safe-MBRL method to solve the environment and even outperform model-free and unconstrained approaches in terms of MAR ($p < 1.75e - 8$), with Safe-MBPO fails completely in reaching any of the goals. At the same time, all the tested algorithms have shown no statistically significant differences from the point of view of the cost.

## 6 Conclusion and Future work

In this study, we proposed GuSS, the first model based planning method using quality-diversity methods for safe reinforcement learning. QD algorithms are methods explicitly designed to provide good exploration. We leverage this property to design a safety-aware planner for GuSS. We demonstrated the necessity of safe exploration in safety-critical settings by comparing our planner to three other (safe and unsafe) planners in a simple toy environment. Only the QD-based safe planner consistently solved the toy environment, achieving high rewards and no cost thanks to maximally exploring the safe space. We further demonstrated on three benchmark environments with soft safety constraints how GuSS compares favorably with state-of-the-art model-free and model-based safe algorithms in terms of the trade off between performance and safety, while requiring minimal computational complexity. Especially on SafeCar-Goal, GuSS is the only method that manage to solve the environment.

In conclusion, Guided Safe Shooting (GuSS), demonstrates promising results in balancing performance and cost in safety-critical reinforcement learning environments. However, the performance of GuSS is dependent on the accuracy of the model used. If the model is wrong, it could easily lead the agent to unsafe states. In future work, we will work on incorporating model uncertainty with QD to inform the agent about the risk of its actions to reduce unsafe behavior during the model learning phase. Additionally, the need to hand-design the behavior space in QD-based algorithms limits the range of applicability. While some works have been proposed to address this issue (Cully, 2019; Paolo et al., 2020; 2023), how to autonomously build such space still remains an open question. Having an algorithm that could learn both a model of the system and a good representation for the behavior space of its planner would likely greatly improve the performance and efficiency of such methods.

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

# A Environments

## A.1 Safe Pendulum

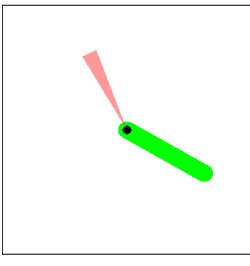

Figure 4: Pendulum upright task.

A safe version of OpenAI's swing up pendulum (Brockman et al., 2016), in which the unsafe region corresponds to the angles in the $[20°, 30°]$ range, shown in red in Fig. 4. The task consists in swinging the pendulum up without crossing the unsafe region. The agent controls the torque applied to the central joint and receives a reward given by $r = -\theta^2 + 0.1\dot{\theta}^2 + 0.001a^2$, where $\theta$ is the angle of the pendulum and $a$ the action generated by the agent. Every time-step in which $\theta \in [20°, 30°]$ leads to a cost penalty of 1. The state observations consists of the tuple $s = (\cos(\theta), \sin(\theta), \dot{\theta})$. Each episode has a length of $T = 200$. We used as behavior descriptor $b = (\theta_{T/2}, \theta_T)$, which is, respectively, the angle of the pendulum at half of the planning trajectory and at the end of it.

## A.2 Safe Acrobot

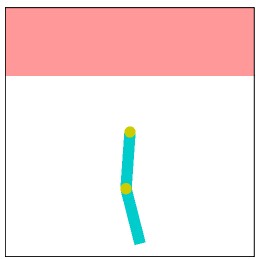

Figure 5: Acrobot upright task.

A safe version of the Acrobot environment (Brockman et al., 2016), shown in Fig. 5. It consists in an underactuated double pendulum in which the agent can control the second joint through discrete torque actions $a = \{-1, 0, 1\}$. The state of the system is observed through six observables $s = (\cos(\theta_0), \sin(\theta_0), \cos(\theta_1), \sin(\theta_1), \dot{\theta}_0, \dot{\theta}_1)$. The reward $r \in [0, 4]$ corresponds to the height of the tip of the double pendulum with respect to the hanging position. The unsafe area corresponds to each point for which the height of the tip of the double pendulum is above 3 with respect to the hanging position, shown in red in Fig. 5. Each episode has a length of $T = 200$ and each time-step spent in the unsafe region leads to a cost penalty of 1.

For this environment, the constraint directly goes against the maximization of the reward. This is similar to many real-world setups in which one performance metric needs to be optimized while being careful not to go out of safety limits. An example of this is an agent controlling the cooling system of a room whose goal is to reduce the total amount of power used while also keeping the temperature under a certain level. The lowest power is used when the temperature is highest, but this would render the room unusable. This means that an equilibrium has to be found between the amount of used power and the temperature of the room, similarly on how the tip of the acrobot has to be as high as possible while still being lower than 3. GuSS' behavior descriptor is $b = (x_{T/2}, y_{T/2}, x_T, y_T)$, corresponding to the end effector (x, y) pose at the middle and at the end of each plan.

## A.3 Safe Car Goal

Introduced in Safety Gym it consists of a two-wheeled robot with differential drive that has to reach a goal area in a position randomly selected on the plane, represented as the green cylinder in Fig. 6.(c). When a goal is reached, another one is spawned in a random location. This repeats until the end of the episode is reached (at 1000 time-steps). On the plane there are multiple unsafe areas, shown as blue circles in Fig. 6.(c), that the robot has to avoid. The placement of these areas is randomly chosen at the beginning of each episode.

The agent can control the robot by setting the wheels speed, with $a \in \mathcal{A} = [-1, 1]$. The observations consists of the data collected from multiple sensors: accelerometer, gyro, velocimeter, magnetometer, a 10 dimensional lidar providing the position of the unsafe areas and the current position of the robot with respect to the goal. This leads to an observation space of size 22.

In the original environment, the cost is computed using the robot position with respect to the position of the different hazards areas. However, as these information are not available in the observations we change the safety function using the Lidar readings and a predefined threshold of 0.9 above which a cost of 1 is incurred. We choose to evaluate the different plans' behavior descriptor as the robot position with respect to the goal at the end of each plan: $b = \sqrt{(x_{robot} - x_{goal})^2 + (y_{robot} - y_{goal})^2}$

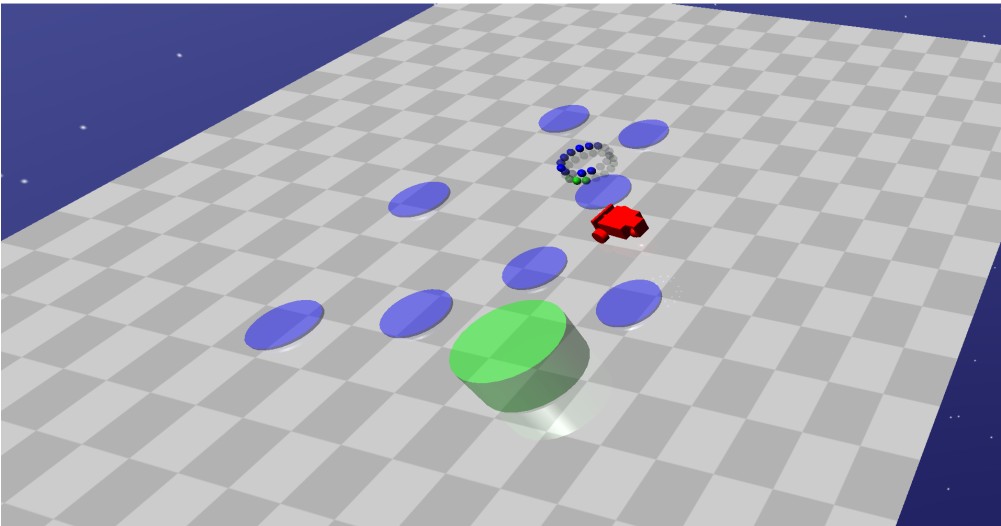

Figure 6: Safety_gym environment

## B Trajectories toy environment

Fig. 7 shows the trajectories followed by the agents tested in Section 5.1 over the 10 seeds. We can see how GuSS can consistently reach the goal without passing over the unsafe areas, while CEM does is going through the unsafe zones. At the same time, both RCEM and S-RS fail to reach the goal and remain stuck around the start due to their limited exploration capabilities.

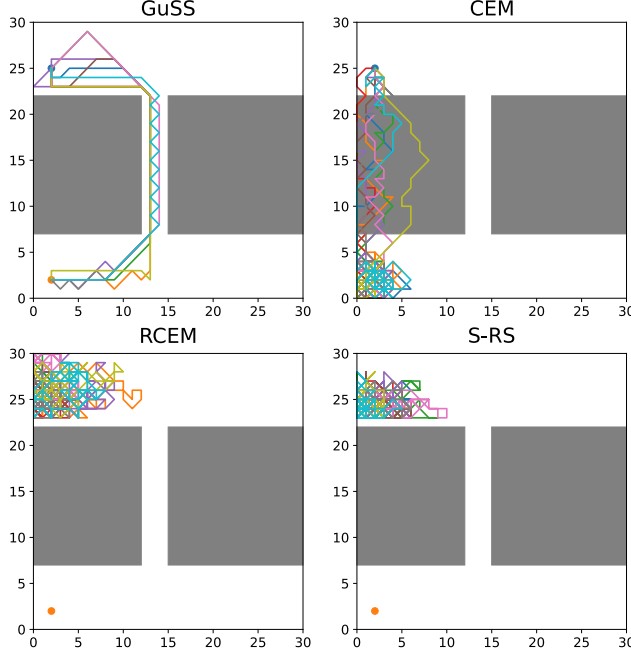

Figure 7: Trajectories followed in the Toy Environment by GuSS and the three tested baselines over 10 seeds.

## C   Optimality

Safe-RL algorithms have to optimize the reward while minimizing the cost. Choosing which algorithm is the best is a multi-objective optimization problem. Fig. 8 shows where each of the methods tested in this paper resides with respect to the MAR score and the p(unsafe). Methods labeled with the same number belong to the same optimality front with respect to the two metrics. A lower label number indicates an higher performance of the method.

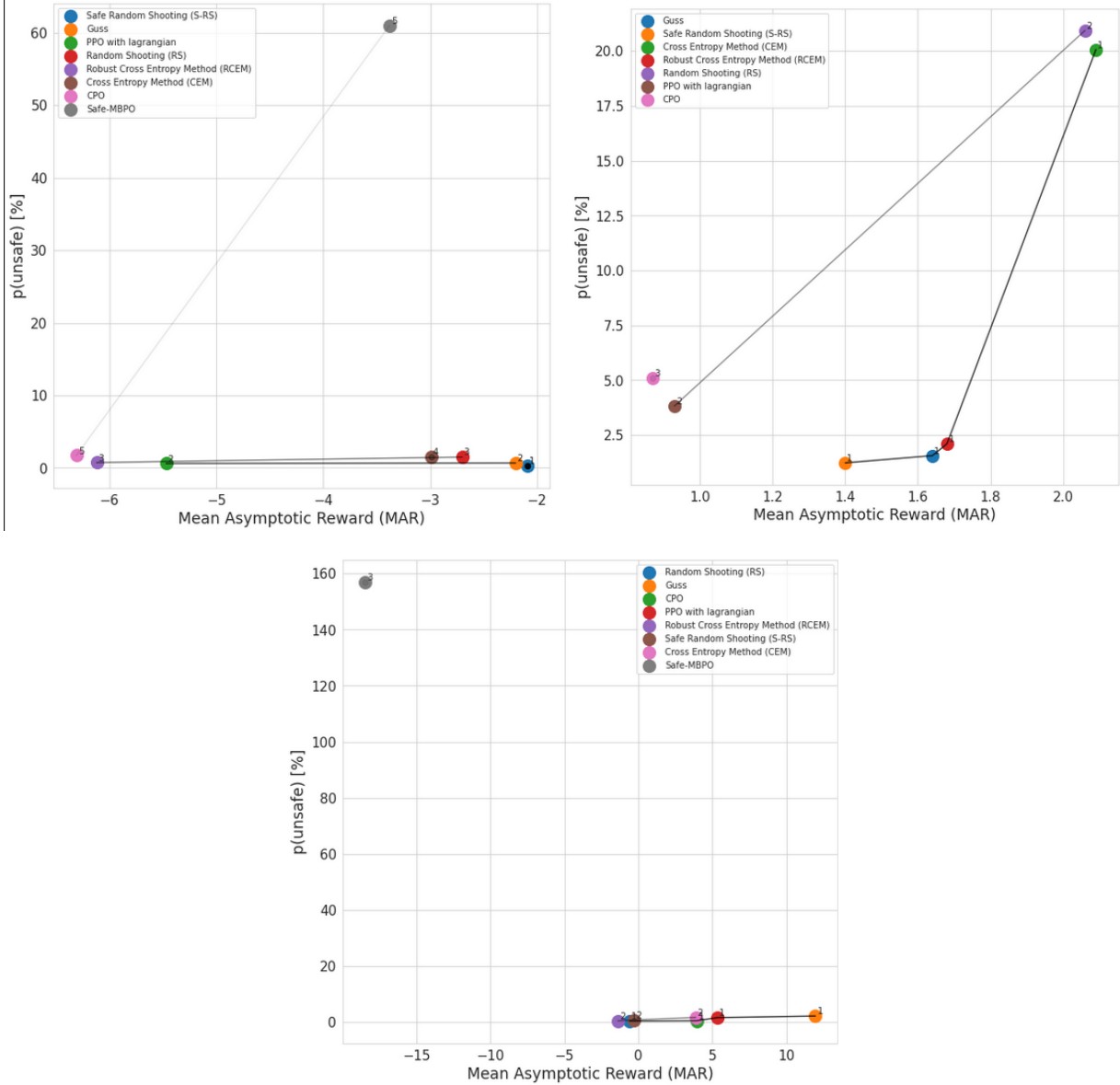

Figure 8: Pareto front for the three environment (top to bottom): Safe-Pendulum, Safe-Acrobot and SafeCar-Goal. Same number belong to the same optimality front with respect to the two metrics. A lower label number indicates an higher performance of the method.

## D   Metrics

In this appendix we discuss the details on how the metrics used to compare the algorithms are defined.

**Mean Asymptotic Reward (MAR)**. Given a trace $\mathcal{T}r$ and a reward $r_t = r(s_t, a_t)$ obtained at each step $t$, we define the mean reward as $R(\mathcal{T}r) = \frac{1}{T}\sum_{t=1}^{T} r_t$. The mean reward in iteration $\tau$ is then $MR(\tau) = R(\mathcal{T}r_t^{(\tau)})$. The measure of asymptotic performance (MAR), is the mean reward in the second half of the epochs (we set $N$ so that the algorithms converge after less than $N/2$ epochs) $MAR = \frac{2}{N}\sum_{\tau=N/2}^{N} MR(\tau)$.

**Mean Reward Convergence Pace (MRCP)**. To assess the speed of convergence, we define the MRCP as the number of steps needed to achieve an environment-specific reward threshold $r_{thr}$. The unit of MRCP$(r_{thr})$ is real-system access steps, to make it invariant to epoch length, and because it better translates the sample efficiency of the different methods.

**Probability percentage of unsafe ($p(\mathbf{unsafe})[\%]$)**. To compare the safety cost of the different algorithms, we compute the probability percentage of being unsafe during each episode as $p(\text{unsafe}) = 100 * \frac{1}{T}\sum_{k=0}^{T} \mathcal{C}_k$ where $T$ is the number of steps per episode. We also compute the *transient* probability percentage $p(\text{unsafe})_{trans}[\%]$ as a measure to evaluate safety at the beginning of the training phase, usually the riskiest part of the training process. It is computed by taking the mean of $p(\text{unsafe})$ on the first 15% training epochs.

# E  Training and hyper-parameters selection

All the training of model-based and model-free methods have been perform in parallel with 6 CPU servers. Each server had 16 Intel(R) Xeon(R) Gold CPU's and 32 gigabytes of RAM.

## E.1  Model Based method

For the Safe Acrobot and Safe Pendulumn environments we used as model $p$ a deterministic deep auto-regressive mixture density network ($DARMDN_{det}$) while for the Safety-gym environment we used the same architecture but without auto-regressivity ($DMDN_{det}$).

Table 2: Model and Agent Hyper-parameters

|  |  | Safe Pendulum | Safe Acrobot | Safe Car Goal |
|---|---|---|---|---|
|  | Model | | | |
| $D(AR)MDN_{det}$ | Optimizer | Adam | Adam | Adam |
|  | Learning rate | 1e-3 | 1e-3 | 1e-3 |
|  | Nb layers | 2 | 2 | 2 |
|  | Neurons per layer | 50 | 50 | 50 |
|  | Nb epochs | 300 | 300 | 300 |
|  | Planning Agents | | | |
| CEM and RCEM | Horizon | 10 | 10 | 30 |
|  | Nb actions sequence | 20 | 20 | 3000 |
|  | Nb elites | 10 | 10 | 12 |
| S-RS and RS | Horizon | 10 | 10 | 30 |
|  | Nb actions sequence | 100 | 100 | 3000 |
| Guss | Horizon | 10 | 10 | 30 |
|  | Nb policies | 100 | 100 | 525 |
|  | Nb initial policies | 25 | 25 | 25 |
|  | Nb policies per iteration | 5 | 5 | 10 |
|  | Behavior space grid size | $50 \times 50$ | $50 \times 50$ | $20 \times 20$ |
|  | Nb policy params. | 26 | 83 | 77 |
|  | Nb policy hidden layers | 1 | 2 | 1 |
|  | Nb policy hidden size | 5 | 5 | 3 |
|  | Policy activation func. | Sigmoid | Sigmoid | Sigmoid |
|  | Mutation parameter $\Sigma$ | 0.05 | 0.05 | 0.05 |

For Safe-MBPO we used their official implementation: https://github.com/gwthomas/Safe-MBPO.

## E.2  Model Free method

All model free algorithms implementation and hyper-parameters were taken from the https://github.com/openai/safety-starter-agents repository.

