# OpenReview forum: "Guided Safe Shooting: model based reinforcement learning with safety constraints"
_TMLR — Rejected by TMLR_

### Review · Reviewer_JZyR · 2024-01-15

**Summary Of Contributions:**

This paper proposes a safety-aware planner based on the QD algorithm MAP-Elite, and tests it on several simulation environments.

**Audience:**

Yes

**Claims And Evidence:**

Yes

**Requested Changes:**

Some questions:
1. Is the use of a squigarrow for representing probabilistic and deterministic maps conventional?
If so, could you please point me to a reference?
You may just define p as that returning probability distribution over the state and consider the delta distribution as deterministic map…

2. page 3 ….is an estimate of the probability of being in an unsafe state… may not make sense; it can be 100 % even thought the MC(Trace)=0.01 for T>100.

3. Page 5  The optimal policy … is the one such that there is no other policy … → isn’t the optimal policy defined by (3)?  Then safety is a hard constraint.  Otherwise one should have a fixed weighting of reward and constraint for discussing optimality.

4. descriptor returning function f is sometimes used as that on trace space and sometimes on the product of policy and trace spaces.  Please unify them.

5. How do you determine the variance of the perturbation to policies?  (in the current form it is fixed to 0.05)

6. Any connection to Thompson sampling?

7. It was not crystal clear in the introduction etc. but this work is not for hard safety constraint case right?  Meaning, we cannot guarantee the probability 1 safety.  And there is no probability guarantee either?   If so, please clearly state it.

8. For the experiment of reaching the goal task; why S-RS visits unsafe region is not clear from the texts.  It says it rejects all the unsafe planner and it also says it is under perfect model.  Then why does it violate the safety?  If this figure is just for a virtually planned trajectory and not an executed one, then please clearly explain so.

9. The subsections of experiments, namely, “exploration” and “environments”, may need to be replaced by a bit more detailed one.  Then for the 5.2 experiments, those are also under perfect models?  If so, it should be clear from the introduction that this work focuses on planning, not learning perspective.

Minor:
1. page 3: citation of Thomas2021, Koller2018, and Vlastelica2021 are in plain text form.
2. Define N^+ (the set of positive integers?)
3. equation 2 is strange; {pi in … and …} looks like it is just a set of conditions.  You may write {pi in Pi: C(pi) <= l_i for all i in {1,2,…,m}}
4. page 4:  … it is possible to recover a safe policy.  → safe state?
5. page 6: DARMDN explanation is a bit hard to parse.  What does p_ell(...|…) mean here?  Please put a bit clearer explanations.
6. page 6: policy set Gamma(0)= [ … ]  → if this is the set, why not write { … } instead?
7. page 6: st+h → s_{t+h}
8. page 9 in figure caption:  one model-based approaches → approach
9. page 9: citation Ray2019 is in plain text form

**Strengths And Weaknesses:**

Strength:
The algorithm itself and the problem this work tackles are very interesting, and the proposed planning method could be one of the useful methods in the literature.

Weakness:
The presentation, especially the notations and mathematical expressions, lacks rigor in some parts, which makes it harder for me to easily understand the contents.  Also, it was less clear where the learning plays role in this work.
That said, if my concerns listed below are addressed, I see this work is a valid proposal of an algorithm for planning.

---

> ### Author Response · Authors · 2024-02-20
> **Requested change and minor modifications**
>
> We want to thank the reviewer for his in depth comments and his help in improving the paper. We update the paper according to the requested changes as follow:
>
> **Is the use of a squigarrow for representing probabilistic and deterministic maps conventional?
> If so, could you please point me to a reference? You may just define p as that returning
> probability distribution over the state and consider the delta distribution as deterministic
> map**
>
> This is not a standard notation however we add a note on the paper to explain our
> notation so it is easier for the reader to understand it. You can find the use of this notation  on this paper Model-based micro-data reinforcement learning: what are the crucial model properties and which model to choose? ICLR 2021.
>
>
> **page 3 ….is an estimate of the probability of being in an unsafe state… may not make sense; it can be 100\% even thought the MC(Trace)=0.01 for $T>100$.**
>
> This a frequentist etimate approach for computing the probability, over the trace $\tau$ we count the number of time we entered  an unsafe state and divide it by the length of the trajectory $T$. To be 100\% it would mean that along all the trajectory the policy violate the safety constraint.
>
> **Page 5 The optimal policy … is the one such that there is no other policy … → isn’t the optimal policy defined by (3)? Then safety is a hard constraint. Otherwise one should have a fixed weighting of reward and constraint for discussing optimality.**
>
> We agree with you that if we follow the definition of equation 3 it is an hard constraint. However, as stated in page 4 we consider here a setup where some violation are acceptable as it is define in Ray, A., Achiam, J. and Amodei, D. (2019) Benchmarking Safe Exploration in Deep Reinforcement Learning. As defined also in their paper, we prioritize safety over reward such that a policy with higher reward but lower safety is consider worse than a policy with lower reward and higher safety.
>
>
> **How do you determine the variance of the perturbation to policies? (in the current form it
> is fixed to 0.05)**
>
> We perform hyper-parameter search to find the value that achieved the best
> result which are the one reported in the paper.
>
> **Any connection to Thompson sampling?**
>
> Could you elaborate on this point?
>
> **It was not crystal clear in the introduction etc. but this work is not for hard safety constraint case right? Meaning, we cannot guarantee the probability 1 safety. And there is no
> probability guarantee either? If so, please clearly state it**
>
> We agree with you following your first remark on the definition of optimal policy, we updated the introduction to reflect this and help better frame the paper.
>
> **For the experiment of reaching the goal task; why S-RS visits unsafe region is not clear from
> the texts. It says it rejects all the unsafe planner and it also says it is under perfect model.
> Then why does it violate the safety? If this figure is just for a virtually planned trajectory
> and not an executed one, then please clearly explain so.**
>
> The only experiment with the perfect model is the one discussed in the exploration section (5.1). In that experiment, S-RS does no perform any violation, but does not reaches the goal either, as it can be seen in appendix B. In the other experiments, safety is violated because the algorithm does not have a perfect model.
>
>
> **The subsections of experiments, namely, “exploration” and “environments”, may need to be
> replaced by a bit more detailed one. Then for the 5.2 experiments, those are also under perfect
> models? If so, it should be clear from the introduction that this work focuses on planning,
> not learning perspective**
>
> Except from the toy environments that we hand designed to test
> the exploration of the different algorithms under a perfect model, all other environments use
> a learned model as describe in section 4.1. We choose to separate the two sections as they
> investigate two different aspect, for the exploration we provide a perfect model and a budget
> to generate safe plan where we show that QD approach is the only one to cover most of the
> search space. Whereas in the environments ( Acrobot, pendulum and SafetyGym) the model
> is learned.
>
> We fix all the minor typos  highlighted by the reviewer.

---

> > ### Comment · Reviewer_JZyR · 2024-02-20
> > **Thank you for the response**
> >
> > Thank you for the response; it seems some of my concerns have been addressed but there are still some concerns about the rigors and clarity.
> >
> > 1. I may have just missed it but I could not find the squigarrow notation in the paper you listed; if you believe this is important notation it is fine, but I think just using the map returning probability distribution p() for clarity.  Delta distr for deterministic map.
> > 2. about "is an estimate of the probability of being in an unsafe state"...  I thought that the safety probability is the probability that the trajectory itself does not experience any unsafe state.  From the current text, it is misleading, so you want to clearly state that the probability of safety is defined to be the proportion of safe states visited along one trajectory.
> > 3. About optimal policy thing; I know what you mean here about soft constraint.  The problem is its rigor of writing.  If you say optimal there must be a well-defined objective, and I naturally see (3) as the objective.  So if you are using different objective than (3) please clearly define that to state optimality.
> > 4. About the parameter 0.05; perhaps you have put it into appendix before and I missed it, but I would appreciate if you clearly write how you did parameter search for this in the main text briefly.  (you may have already put it though)
> > 5. Thompson sampling is sampling based way for doing exploration-exploitation tradeoff; if you are investigating exploration capability you want to discuss some connections to those existing techniques.
> > 6. I still don't get it about the safe planner percentage.  Figure 2 is for perfect model setting or not?  If it is for perfect model setting why (b) shows unsafe plans as well?
> > 7. What I meant for "exploration", "environments" was that it is better to put more clear subsection title to them, not about the contents.  You can say for example, "Testing exploration capability under perfect model settings" or so.  Otherwise it is hard to parse what the subsection is for.
> >
> > Again, I see the value of this technique itself; but the major concern is the clarity and rigor of notation etc. which can mislead readers having different thinking process.  I am now getting the idea better but I would like to see the above addressed as well.

---

> > > ### Author Response · Authors · 2024-02-21
> > > **Revised version**
> > >
> > > Thank you for your quick response and clarifications, we updated the paper according to your comments:
> > >
> > > **1. I may have just missed it but I could not find the squigarrow notation in the paper you listed; if you believe this is important notation it is fine, but I think just using the map returning probability distribution p() for clarity. Delta distr for deterministic map.**
> > >
> > > The definition is  in page 4, in the footnote where the notation first appear.
> > >
> > > **2. about "is an estimate of the probability of being in an unsafe state"... I thought that the safety probability is the probability that the trajectory itself does not experience any unsafe state. From the current text, it is misleading, so you want to clearly state that the probability of safety is defined to be the proportion of safe states visited along one trajectory.**
> > >
> > > We updated the sentence as you suggested: *... is an estimate of the probability of unsafe states visited along one trajectory*.
> > >
> > > **3. About optimal policy thing; I know what you mean here about soft constraint. The problem is its rigor of writing. If you say optimal there must be a well-defined objective, and I naturally see (3) as the objective. So if you are using different objective than (3) please clearly define that to state optimality.**
> > >
> > > We updated the sentence in section 3.2 with the equation 3: "*With a good planner that spans the search space we can
> > > expect to have Πc ⊆ ΠM , then the optimal policy π∗ can be found by taking the objective of equation (3)
> > > such that there is no other policy π′ ∈ ΠM with a lower MC(T ) and higher MR(T ) (Ray et al., 2019).*.
> > >
> > > **4. About the parameter 0.05; perhaps you have put it into appendix before and I missed it, but I would appreciate if you clearly write how you did parameter search for this in the main text briefly. (you may have already put it though)**
> > >
> > > We added a sentence as suggested in section 5.2: *A detailed description of the environments
> > > and the hyperparameters optimized by grid search are provided in Appendix A.*
> > >
> > > **5. Thompson sampling is sampling based way for doing exploration-exploitation trade-off; if you are investigating exploration capability you want to discuss some connections to those existing techniques.**
> > >
> > > The difference with Thompson sampling is that we do not have any trade off. This is thanks to the discretized behavior space. The algorithm explores, and if it finds similar policies, it keep the best one. It does not have to concern itself with any trade-off
> > >
> > > **6. I still don't get it about the safe planner percentage. Figure 2 is for perfect model setting or not? If it is for perfect model setting why (b) shows unsafe plans as well?**
> > >
> > > The plans are not what are executed, but what are generated during the search in the perfect model before evaluating their cost and their reward. A planner can generate 100 plans before selecting which one to execute. If these 100 plans are all unsafe, the percentage of unsafeness is 100. The planner then selects the one with the lowest cost and highest reward. This metrics was choose to show how well the different planners explore the safe space and we can see that our propose algorithm is the one that generate the most safe plans.
> > >
> > > **7. What I meant for "exploration", "environments" was that it is better to put more clear subsection title to them, not about the contents. You can say for example, "Testing exploration capability under perfect model settings" or so. Otherwise it is hard to parse what the subsection is for.**
> > >
> > > We update the section name's to help the reading as suggested: Exploration -> Safe exploration with known model of environment and Environments -> Safe exploration with world model learning.
> > >
> > > We hope this changes address reviewer comments and concerns.

---

> ### Comment · Reviewer_JZyR · 2024-02-21
> **Thank you for the response again**
>
> Thank you for the response;
>
> 1-- Yes I know, I mean you listed "Model-based micro-data reinforcement learning: what are the crucial model properties and which model to choose? " as a paper that uses this notation.  But I could not find the use in this paper.  Anyway, if you believe this use is fine, it is ok.
>
> 3-- still, it does not seem rigorous.  if the objective is (3), the optimal policy is the one achieving highest MR while satisfying hard constraint.  The writing "such that there is no other policy .... with a lower... and higher..." is ambiguous; are you creating a surrogate objective to (3)?  If so, please describe that and write the true objective in equation form.  If you are really talking about the optimal policy w.r.t. (3), then the writing seems weird.
>
> 6-- yes that makes sense, then please write so clearly in the text.
>
> For 5., you don't necessarily describe it in depth in the paper, but I am a bit confused that you say there is no trade-off.  Exploration is one of the keys of this work, but it is not just random exploration, meaning you are also aiming for achieving higher reward.  In that sense, it is doing some tradeoff and if this tradeoff is optimal in some sense that might be a good algorithm.  So I thought your techniques are naturally balancing exploration and exploitation and wanted to see some discussions around here.  But it may not be necessary for this work.

---

> > ### Author Response · Authors · 2024-02-28
> > **Update response**
> >
> > Thank you for your response and help, we addressed your concerns as follow:
> >
> > 3. We re-write the objective following your comments (please see equation 4).
> >
> > 6. We update caption of Fig 2 especially sub-caption *b* to better explained that this is the generated plans and not the executed one.
> >
> > 5. We agree with you, to better clarify our point QD-ME method do this trade-off implicitly as these family of algorithms maintain a population of candidate solutions (also known as individuals or solutions), and over multiple generations, they evolve this population to find a diverse set of solutions that represent the trade-offs between the conflicting objectives.
> >
> > We hope this revisions address reviewers concerns.

---

> > > ### Comment · Reviewer_JZyR · 2024-02-28
> > > **Thank you**
> > >
> > > The author(s) have addressed my concerns.

---

### Review · Reviewer_ZvND · 2024-01-17

**Summary Of Contributions:**

The manuscript presents a model-based reinforcement learning (MBRL) algorithm that essentially builds on an evolutionary strategy for generating candidate policies. In addition to a reward formulation, also unsafe states are incorporated by formulating them as a cost, which shall be minimized corresponding to the goal to stay away from unsafe regions. The proposed algorithm combines existing model learning techniques with existing evolutionary search strategies.  Key aspect is the simultaneous consideration of performance and safety in the above sense, in particular when generating new candidate policies. The proposed method GuSS is evaluated in three simulation environment, where it shows a lower portion of visits to unsafe states compared to other algorithms.

**Audience:**

No

**Broader Impact Concerns:**

In my opinion it is fine, and does not need to be specifically addressed here.

**Claims And Evidence:**

No

**Requested Changes:**

Critical ones correspond to the above mentioned weaknesses (W1) to (W7)

**Strengths And Weaknesses:**

## Strengths

(S1) Proposed algorithm shows competitive or superior performance on the chosen simulation benchmarks in terms of performance and cautious/safe behavior

(S2) Relevant problem (even if states as "cautious" exploration instead of "safe", see below)



## Weaknesses

(W1) Overstatement regarding "safety"

My main concern is that I doubt whether it is justified to call the proposed method "safe". Safety is not treated as an actual safety constraint, but as another cost in the RL problem, which one seeks to minimize.  Hence, one seeks to have little safety violations, but based on the formulation, there is no way to have any guarantee. Furthermore, the entire safety consideration hinges on the model that is learned from data. However, there is no way to quantify model quality (with the used methods), and thus, no way to know whether the model is actually good or bad. Finally, the empirical results actually do show some visits of unsafe states.

IMHO, in order to claim any safety **guarantee**, one needs theoretical statements.  However, the manuscript does not have any, and I also don't see how any guarantees could be obtained for the type of evolutionary methods and model learning methods proposed here. To be safe, however, guarantees are needed (as the manuscript also states, see, e.g., the abstract).

Overall, I find it misleading (if not dangerous) to call the approach "safe" and talk about safety guarantees in this context. As is visible in the empirical results, the algorithm has value in being effective in avoiding unwanted regions and states. For this type of behavior, **cautious** is the better word, but IMHO, "safety" should not be claimed.  I thus suggest to frame the method and whole paper as an approach for cautious exploration.



(W2) Discussion of related work is insufficient to support the novelty and relevance of the contributions.

The innovation over the related work, and thus the relevance of this work, is not very clearly presented and spelled out.  After reading the related work section (Sec. 2), I could not pin point what is better / the improvement over the SOTA.  In particular:

* The discussion of related work is of limited insight regarding this work and its contributions.  It reads essentially as a list of some works on safe RL, but it is missing statements on how the mentioned works related to this work (like what is similar, what is different).  As is, I did not understand the potential novelty or advantage of the proposed method.
* The manuscript refers to Amodei et al (2016) as a general reference on AI safety problems.  There are more recent overview-type references / surveys, for example, *Brunke et al., "Safe learning in robotics: From learning-based control to safe reinforcement learning", Annual Review of Control, Robotics, and Autonomous Systems, 2021*. The authors should include more recent overview articles, and use the discussion and references therein in order to put their work into context.
* There are no references from 2023 and only few from 2022.  This seems surprising as safety in RL and learning-based control is a hot topic.



(W3) Statement of contributions

* It seems that the two contributions listed at the end of the introduction are actually one contribution.  Proposing QD and ME (Contribution 1) actually results in the GuSS algorithm (Contribution 2).  Correct me if I'm wrong, but I didn't see these as two separate contributions.
* Furthermore, I don't find this (or these) contributions to be very precisely stated. An algorithm is proposed, which aims to avoid unsafe states. This could be stated for many other safe learning algorithms likewise -- what is different / the special feature or idea?  Further, how is the algorithm superior to existing work?  This is not clear from the list of contributions.

* From my understanding, none of the components in the proposed algorithm GuSS are new. For model learning, existing components are used (Sec. 4.1). Likewise, the evolutionary policy search seems to use existing methods as components (Sec. 4.2).  If this is correct, I find the contribution minor, which is, however, not the main criterion in TMLR review.  Nonetheless, it should be transparently stated already in the introduction that existing components (and which) are combined. If my understanding is incorrect, it needs to be clearly and explicitly stated what components are known and which are new.



(W4) Some arguments in the introduction regarding key aspects of the paper are simplistic and not well supported (page 2).

* Page 2: In what sense does the used model learning technique ensure "for minimal real-system interactions"?  This should be clarified as there are a gazillion of model learning techniques and it is unclear (based on the description) why the proposed one should be superior.

* "fewer interactions with the real-system mean less chance of entering unsafe states". This seems simplistic.  The ideal goal should be to design a mechanism that can safely explore (e.g., as done in methods on Safe Bayesian Optimization); if this is in place, more data won't harm safety. In fact, generating MORE data by carefully expanding a safe set in small increments can be a better strategy from the safety point of view than going for maximal information with as few samples as possible.  Further, taking the above to the extreme, one could not sample at all and would be 100% safe.  To clarify my point, I see the usefulness of needing only few samples, but the implication "small data => better safety" is too simplistic IMHO.
* Page 2, "by learning a model of the system, this allows flexibility and safety guarantees as by using the model we can anticipate unsafe actions before they occur".  I don't buy this argument that learning a model (by itself) can give you safety **guarantees**.  The guarantees can only be as good as the model. As the model itself is learned, there is no guarantees a priori, and also during learning guarantees can be given (if at all) for regions where sufficient data has been seen.
* What are the main characteristics of the proposed planner that is based on QD and ME?  These concepts are not explained in the introduction, yet, as this is a claimed contribution of this work, I find it important to explain the key ideas already in the introduction.  As is, I was unable to see and understand the contribution from the introduction.



(W5) Key methods/concepts used in the proposed algorithm are not sufficiently introduced.

* Sec. 3.3:  I do not understand what the behavior space \mathcal{B} is and how it is "hand-designed".  Given that QD is a key proposal of this manuscript, this description is insufficient.
* Further, even though it is not well explained, I seem to understand that the hand-designed \mathcal{B} (and later f(.)) is critical in evaluating policies.  Then it seems even more important to properly explain this, as this might somewhat defeat the purpose of RL if the designer needs to judge what policy/trajectory is good.
* Same, the ME algorithm is not explained either.



(W6) Experiments: better discussion of competitor algorithms

* Sec. 5.1: Why did the authors choose to compare to CEM, RCEM, and S-RS?  No motivation or explanation is given.  What is the state of the art algorithm for such safe planning problems?
* The comparison on the OpenAI gym environments includes a better and wider selection of algorithms.  Still, it would be good if the authors clarified  (e.g., by relating to the related work description) what is the state of the art in safe MBRL.



(W7) Experimental evaluations

* What does p(unsafe) mean precisely (Table 1, Figure 3)?  I guess it is the portion of unsafe state visits and thus an empirical **estimate** of being safe.  As this is the key aspect of the paper, this needs to be explained in the main paper, i.e., the precise computation and its interpretation.

* Main plots in Figure 3 need to be improved

  * 6 random seeds seems a rather low number.
  * The plots show the means only.  Variations from the 6 runs should also be visualized, as is common in RL.
  * Figures are hard to read. Font size is too small, and figures too dense.
  * I do not understand what the red dashed line means.



## Minor comments and suggestions

* It would be helpful to understand, already in the introduction, which environment the method is being evaluated to get an understanding on how comprehensive the evaluation actually is.
* What is called "Model-based RL with MPC" in Sec. 3.2 is classically known as MBRL with decision-time planning (see Sutton/Barto).  IMO, it would be better to refer to established terminology.  Especially as "MPC" is not explained in the manuscript.
* There are several typos.  **Examples**: p. 3, "Thomas2021", "Vlastelica2021", "Koller2018"; p. 4, "i.e policy"
* What does "\mathcal{T}r" (e.g., Alg. 1, Sec. 3.2) mean versus "\mathcal{T}" ??
* For some references, only the arxiv reference is given, while also another reference (journal/conference) exists.  The authors should check for all references.

---

> ### Author Response · Authors · 2024-02-20
> **Requested changes and clarifications**
>
> We want to thanks the reviewer for his thoughtful review, we list hereafter all the requested changes raise by the reviwer and how we address them:
>
>
> **(W1) Overstatement regarding ”safety”**
>
> We agree with you on the possible confusion of the use of the word safety and the need for
> theoretical guarantees. We use the word safety as it is commonly use in literature [1], we update
> the intro to emphasis on the fact that we are not addressing zero violation safety ( safety level III) which is the terminology used in Safe-RL that aligned with your comments. We update the intro and
> abstract to better indicate that we are not proposing a zero-violation algorithm but address soft safety constraints (safety level I [1]).
>
> 1.  From learning-based control to safe reinforcement learning”, Annual Review of Control, Robotics, and Autonomous Systems, 2021
>
> **(W2) Discussion of related work is insufficient to support the novelty and relevance of the
> contributions.**
>
> We have revised the section on related work to better align our method with existing strategies, particularly Model-Based Reinforcement Learning (MBRL) with Cross-Entropy Method (CEM) planning, as it is the closet approach to our propose method.
>
> **(W3) Statement of contributions**
>
> We updated the introduction to reflect your comments, especially we better highlight our
> contribution. As you point out the contribution of the paper build on existing work, the novelty lies in how we adapted it for Safe-RL framework. We updated the Related work section to better
> position our paper with respect to existing work. The special feature and idea is the use of quality diversity algorithm to generate safe plan and its adaptation to consider reward and cost to evolve the population.
>
> **(W4) Some arguments in the introduction regarding key aspects of the paper are simplistic
> and not well supported (page 2)**
>
> We agree with the reviewer that some sentences where simplistic and not supported by our experiments. We rewrote the introduction addressing the different points raised by the reviewer mainly by rephrasing sentences that were too simplistic.
> We described in more detail the characteristics of QD as a planner and better explained our contribution.
>
> **(W5) Key methods/concepts used in the proposed algorithm are not sufficiently introduced.**
>
> We updated the Quality diversity and MAP-Elites section to better explain the algorithms and added detailed pseudocode, also highlighting the differences between our proposed variation of the planner and the vanilla ME algorithm.
>
> **Sec. 3.3: I do not understand what the behavior space \mathcal{B} is and how it is "hand-designed". Given that QD is a key proposal of this manuscript, this description is insufficient. **
> The behaviour space in QD algorithms is a representation of the diversity of solutions, this is a limitation of QD approach as it has to be tailored to the target application, even if the autonomous learning of the behavior space has been approached in multiple works that we cited in section 4.
> We have also presented some examples of what the behavior space could be in some situations like maze navigation or robotic walking tasks.
>
> We add an Appendix where we show the different behaviour that we use for each environment, as stated in the conclusion an extension of this work could focus on building autonomously the  behaviour space.
>
> **(W6) Experiments: better discussion of competitor algorithms**
>
> We choose to compare to CEM, RCEM and S-RS as those are techniques that rely on
> sampling plans using a learned model. By comparing with those algorithms we could show the superior exploration
> of QD, which from our experimental results is an important aspect to solve environments under safety constraints.
> We update also the Results section to better explained our choice for the baselines.
>
> **W7) Experimental evaluations**
>
> We change the figure resolution for better reading, and updated the caption to better explain the red dash line which represents the safety cost of a random agent. We choose to not add the std on the figure as it will make it harder to read but we added it in Table 1. We further add in the appendix a detailed description of the metrics.`
>
> **It would be helpful to understand, already in the introduction, which environment the method is being evaluated to get an understanding on how comprehensive the evaluation actually is. **
> We mentioned in the introductions the environments on which our approach has been tested
>
> **What is called "Model-based RL with MPC" in Sec. 3.2 is classically known as MBRL with decision-time planning (see Sutton/Barto). IMO, it would be better to refer to established terminology. Especially as "MPC" is not explained in the manuscript.**
> We changed the name of the section to reflect that.
>
> **What does "\mathcal{T}r" (e.g., Alg. 1, Sec. 3.2) mean versus "\mathcal{T}" ??**
> We explained this more clearly
>
> We fix all the minor comments raised by the reviewer, typos and reference mistake.

---

> > ### Comment · Reviewer_ZvND · 2024-03-03
> > **Response to authors' response**
> >
> > I appreciate the response by the authors and the upload of a revised version. Some aspects have been improved w.r.t. my comments. For example, the discussion of "safety" is more balanced now in the introduction. Still, I find the notion of safety rather weak and no theoretical guarantees are included (even about the notion of "soft safety"). That being said, it was hard to track down all exact details of what changes the authors have made and how they have responded to previous comments. In part, this is because the authors didn't upload a version of the manuscript with changes highlighted, and in part, because not all detailed points of my review were answered explicitly. For any potential revision, I would prefer if the authors could provide such detailed documentation, also in appreciation of the reviewers' time.
> >
> > While the paper has improved, I could not see sufficient evidence that the current version is fully satisfactory yet for publication. But I will be happy to take the improvement into account for my rating.

---

> > > ### Author Response · Authors · 2024-03-04
> > > **Response to reviewer**
> > >
> > > Thank you for your feedback. We apologize for any inconvenience caused by the tracking of changes. While we made efforts to link every modification in the paper to the reviewer’s comments, we recognize that directly providing these changes in the revised version would have facilitate the revision of the paper.
> > >
> > > Regarding theoretical guarantees, we acknowledge their absence in the current version. However, our approach involves experimentation and metric design to gain practical insights into the performance of our methods, particularly in scenarios involving ‘soft safety.’ Notably, our experiments with a known world model highlight the superiority of QD-ME methods in covering the safe state space.
> > >
> > > Future work, following reviewer comments, could focus on deriving theoretical guarantees under a learned model and exploring how we can incorporate the uncertainty of the model into trajectory optimization search. We view this paper as primarily experimental, and we believe it can be useful for the community. Specifically, our results demonstrate the promise of combining QD-ME algorithms for trajectory optimization in model-based RL, especially in multi-objective settings.
> > >
> > > We thank you for your general comments and your assistance in refining the paper for publication.

---

### Review · Reviewer_LkF8 · 2024-02-06

**Summary Of Contributions:**

This paper proposes a model-based RL approach by utilizing QD methods to solve problems with safety concerns. Experiments are evaluated on two Gym tasks and one safety-gym task.

**Audience:**

Yes

**Broader Impact Concerns:**

Not applicable.

**Claims And Evidence:**

No

**Requested Changes:**

1) The technical details need to be clarified:

   a) What's the definition of the cost function? Is $b$ pre-defined? If so, how do we define $b$ given different environments? What's the difference between $\bar{b}$ and $b$?

   b)  In Section 4.1, could you explain the definition of the autoregressive deep neural nets $p_l$? Why can a part of state variables predict the remaining part $s_{t+1}^l$? Should the $l$ start from 0?

   c) How does the planning achieve safety?

   d) If a pool of random policies is used to do the planning, how can it be safe and high-reward? How to update these policies? Only by perturbing by adding noise?

2) The references should be carefully checked and clarified, for example, in the Related Work section, Thomas2021 citation is not shown normally.

3) A discussion about how to apply this method to a more practical domain could be helpful to support the claim.

**Strengths And Weaknesses:**

Strengths:

1) The focus of this paper is appealing to the community, as model-based RL is a good way to solve real-world problems with safety concerns.

2) The proposed method is evaluated with extensive comparison baselines.

Weaknesses:

1) The reviewer finds the paper hard to follow, especially with the methodology part:

   a) The reviewer is confused about how the planning achieves safety.

   b) There are a lot of definitions that are not well explained, which makes the methodology confusing.

   c) Some formulas have flaws and need to be clarified.

2) The paper is over-claimed. No results on real problems support the claim.

---

> ### Author Response · Authors · 2024-02-20
> **Requested changes**
>
> We want to thank the reviewer for his comments and time, please find our response to the requested changes:
>
> **a.1) What's the definition of the cost function?**
>
>  The definition of the cost function is given in Sec 3.1. We've repeated that in sec 4 to make it more clear.
>
> **a.2) Is $b$ pre-defined? If so, how do we define $b$
>  given different environments?**
>
> $b$ is predefined depending on the environment. It encodes some features about what is important in the environment and to solve the task. We have added a sentence in sec 3.3 better explaining this and added some examples. There are works that automatically learn b that can be used, but in our paper we don’t use them to better analyse the safety performance of our proposed planner. We further include in the appendix all the different behaviour function for each environment.
>
> **a.2) What's the difference between $\bar{b}$ and $b$?**
>
> The barred version is the discretized one used by MAP-Elites, as explained at the end of sec 3.3. We made the distinction clearer in the text.
>
> **b) In Section 4.1, could you explain the definition of the autoregressive deep neural nets $p_l$? Why can a part of state variables predict the remaining part $s^l_{s_{t+1}}$ ? Should $l$ the  start from 0?**
>
> There was an error in the indexing, we cleared that out, now it’s easier to understand how the autoregressiveness works. The assumption with autoregressive NN is that some features can be informative for the predictions of other features. E.g. feeding the NN the predicted current position, together with the previous position, can help in predicting the current velocity.
>
> **c) How does the planning achieve safety?**
>
> It does so by rejection sampling, by selecting, among
> the discovered policies, the one that has the lowest cost and the highest reward.
> We made that more explicit in section 4 and in the algorithms.
>
> **If a pool of random policies is used to do the planning, how can it be safe and high-reward? How to update these policies? Only by perturbing by adding noise?**
>
> The policies are random just at the beginning of the search process, then they are refined through random perturbations and selections to generate both more diverse and better performing policies. Random perturbation is the standard way of generating new neural network policies in an evolution based algorithm.
>
>
> **The references should be carefully checked and clarified, for example, in the Related Work section, Thomas2021 citation is not shown normally.**
>
> We cleaned and reformatted all citations.
>
> **A discussion about how to apply this method to a more practical domain could be helpful to support the claim.**
>
> We added examples in the introduction in which it is clear where such an approach can be useful
>
> We hope that the changes address the reviewer concerns, we also rewrote the introduction and methodology sections, following other reviewers' comments, to make the paper easier to follow and to better state our contribution.

---

> > ### Comment · Reviewer_LkF8 · 2024-03-05
> > **Thanks for the response**
> >
> > Thanks for the response. I think the clarification addressed my concern about methodology. I am still worried about whether the simple evolution method is appropriate to guarantee safe planning. Since there is no evidence to show random initialization and random perturbation can generate high-quality policies.  As to the example in the introduction, it is still unclear to me how this approach could be applied to such a domain.

---

> > > ### Author Response · Authors · 2024-03-05
> > > **Response to Reviewer comment**
> > >
> > > What guarantees the high-quality of the solutions is the selection we apply on the generated policies.
> > > Performing selection that pushes at the same time towards diversity and quality allows to almost completely cover the search space (some algorithms of the same family of the evolution method showed here have been shown to obtain uniform coverage of the search space https://hal.science/hal-02561846/document ).
> > > Thanks to this property, that allows much greater exploration than the other planners we considered, we can then select the policies that satisfy our criteria of performance and safety.
> > >
> > > Applying our algorithm in the scenario presented in the intro is straightforward. It can be done either by providing a pretrained model on precollected data, and then use the planner to select the best temperature setup given the current state, or by learning the model on the flight.
> > > It is important to plan in such settings given the slow changes in temperatures in systems like this, when having the goal of minimizing the risk for the servers (so swiftly reacting to predicted increases in computational demand) while reducing energy comsumption (for example by not forcing the fans of the cooler to the max (strategy that would rapidly decrease the temperature, but at the cost of high energy consumption)).

---

### Decision · Action_Editor_UuaN · 2024-03-20

**Recommendation:** Reject

**Comment:**

The usefulness of the approach, specifically safety guarantee claim, should be properly established in the paper. While the revision addressed some of the concerns, the reviewers remain mostly unconvinced. The authors should improve the paper by addressing the reviewers' points for future submission. Specifically, if it's "soft safely" instead of "guaranteed safety", please rephrase your claims and clarify the practical value of such. In addition, please respond to Reviewer ZvND's points more directly as suggested in their review and included a version with highlighted changes.

**Audience:**

Yes.

**Claims And Evidence:**

All the reviewers think the authors are working on an interesting problem. However, the reviewers all have concerns about the approach and the claims made in the work. One important issue raised by the reviewers (especially Reviewer LkF8 and ZvND) is that the approach that has a data-drive nature does not truly guarantee safety.

In the recommendation, Reviewer LkF8 mentioned "While some concerns remained unsolved: 1) the motivation of safe planning is not well supported. The reviewer is not convinced that random policies with random perturbations can guarantee safe planning. 2) It's still unclear how the method could be useful in safety-concerned practical domains." Reviewer ZvND on the other hand argues that a "theoretical" approach is needed to guarantee safety. The paper suffers from the lack of clarity in the current manuscript. While Reviewer JZyR became satisfied with the revision and the discussion, Reviewer LkF8 and ZvND remained unconvinced.

In conclusion, the paper will benefit from a better presentation of the work. Particularly, the safety guarantee should be carefully or properly claimed and evaluated.

**Resubmission Of Major Revision:**

The authors may consider submitting a major revision at a later time.